# Dextran-based T-cell expansion nanoparticles for manufacturing CAR T cells with augmented efficacy

Tao Zheng [1,4], Keerthana Ramanathan [1,4], Maria Ormhøj[1,4], Mikkel Rasmus Hansen[1], Hólmfridur Rósa Halldórsdóttir[1], Hanxi Li [1], Kamilla Kjærgaard Munk [1], Carlos Rodriguez-Pardo [1], Rasmus Ulslev Wegener Friis[1], Islam Seder[1], Peter M. H. Heegaard [1], Klaus Qvortrup [2], Hinrich Abken[3], Yi Sun [1] ✉ & Sine Reker Hadrup [1] ✉

Adoptive T cell therapy using chimeric antigen receptor (CAR) engineered T cells is currently being explored in multiple cancer types beyond leukemia/lymphoma. A key step in CAR-T cell manufacturing is the activation and expansion of T cells, which facilitates viral transduction, however, may hamper T cell fitness and reduce in vivo persistence. "T-Expand" is developed for T cell activation and expansion, comprising dextran-based nanoparticles conjugated with anti-CD3 and anti-CD28 antibodies. The nanoparticles trigger robust polyclonal expansion of human T cells with efficiency in the range of commercial microbeads (Dynabeads™). Engineered in the presence of T-Expand, CD19 CAR T cells display enhanced proliferative capacity, cytotoxicity and persistence in vitro, and furthermore, exhibit potent anti-lymphoma activity in mouse models, resulting in complete tumor clearance at one fourth of the CAR T cell dose. Importantly, T-Expand is biocompatible with no observed toxicity, circumventing removal steps after T cell expansion compared to Dynabeads™. As a biocompatible T cell expansion platform, T-Expand simplifies the manufacturing process while enhancing T cell persistence and functionality, and thereby holds promise for increasing clinical efficacy of CAR T cell therapy.

T cells genetically modified to express a chimeric antigen receptor (CAR) has shown tremendous potential against a variety of hematological malignancies[1,2]. Despite the clinical success, a substantial number of patients do not gain long-term treatment benefit, potentially owing to T cell intrinsic factors that lead to lack of functional capacity and in vivo persistence[3,4]. While the genetic modification of T cells requires potent T cell activation to enter a proliferative state susceptible to gene modification, the expansion phase imprints the functional characteristics on the resulting CAR T cells[5].

Current conventional CAR T-cell expansion platforms are based on immobilized anti-CD3 and anti-CD28 antibodies on magnetic microbeads (Dynabeads™) or on a polymer matrix (TransAct™)[6,7]. However, these artificial antigen-presenting cell (aAPC) platforms can lead to overstimulation of T cells causing cell exhaustion and ultimately result in lack of persistence after infusion[8]. Alternative micro-scale aAPC platforms such as lipid-coated mesoporous silica scaffolds promote T cell expansion with demonstrated anti-tumor activity in vivo[9,10]. Recently, aAPC platforms based on graphene oxide flakes

[1]Experimental and Translational Immunology Section, Department for Health Technology, Technical University of Denmark, Lyngby, Denmark. [2]Core Facility for Integrated Microscopy, Faculty of Health and Medical Sciences, University of Copenhagen, Copenhagen, Denmark. [3]Division of Genetic Immunotherapy, Leibniz Institute for Immunotherapy, Regensburg, Germany. [4]These authors contributed equally: Tao Zheng, Keerthana Ramanathan, Maria Ormhøj. ✉e-mail: suyi@dtu.dk; sirha@dtu.dk

and viscoelastic microgel have been reported[11,12]. These platforms could enhance expansion kinetics as compared to conventional platforms. However, graphene oxide based aAPCs possess a technical challenge when isolating expanded T cell products from scaffolds, which may prevent generation of therapeutic CAR T cell products[9]. Furthermore, the microgel-based aAPC platforms have difficulty reproducing and maintaining mechanical properties when used at large scale, thus posing challenges for scale-up manufacturing[13].

To address these limitations, we designed a unique dextran-based nano-aAPC, called T-Expand, comprising dextran nanoparticles (NPs) decorated with anti-CD3 and anti-CD28 antibody on its surface. Dextran presents notable advantages, such as high density of modifiable functional groups, which enables conjugation of high amount of T cell stimulatory molecules. We hypothesize that enhancing the surface density of these stimulatory molecules could effectively compensate for the reduced physical contact area inherent to nanoscale aAPCs, thereby maintaining robust T-cell expansion. Meanwhile, we hypothesize that the avidity between T-Expand and T cells is not as strong as that between microbeads and T cells. Upon T-cell activation, T-Expand is either internalized by T cells via the TCR or detached from the T-cell surface, which may help prevent T-cell exhaustion by avoiding prolonged stimulation. In addition, T-Expand is biocompatible, thereby avoiding the extra separation step and therefore suitable for large-scale production.

We demonstrate that T-Expand offers an alternative approach to enhance the quality of CAR T-cell products and simplify the CAR T-cell manufacturing process with beneficial functionality. CAR T cells expanded with T-Expand exhibit significantly improved persistence and therapeutic efficacy in both in vitro and in vivo settings. Specifically, CD19 CAR T cells expanded using T-Expand display reduced exhaustion markers compared to those generated using Dynabeads™. Additionally, T-Expand-produced CAR T cells are applicable for long-term tumor clearance, underscoring their potential clinical advantages over conventional aAPC platforms.

## Results

### Characterization of the T-Expand

Dextran was selected as the material for T-Expand due to its biodegradability, biocompatibility, abundant functional groups and ease of chemical modification[14–16]. Herein, we modified the dextran polymer with multiple copies of azido groups, which enabled flexible conjugation of antibodies in the aqueous phase via bio-orthogonal approaches such as azido-alkyne cycloaddition and Staudinger ligation (Fig. 1a). Firstly, the azido linker was activated using carbonyldiimidazole (CDI) (Supplementary Figs. 1–4). Subsequently, the CDI-functionalized azido linker was reacted with the dextran to produce azido-modified dextran polymer (Supplementary Figs. 5, 6), which was then acetylated to form the acid-sensitive azido-dextran polymer (Supplementary Fig. 7). The resulting hydrophobic azido-dextran NPs ("naked" NPs) were prepared by double emulsion technology. Measurement by NP tracking analysis (NTA) system showed that the naked NPs had a homogenous size distribution of about 150 nm (Supplementary Fig. 8a, b).

The hydrophobicity of the azido linker plays an important role in determining the orientation of the azido groups during the water/oil/water emulsion formation[17]. To investigate the effect of the linker on the amount of azido groups on the surface, we selected two types of linkers, hydrophobic azido linker and amphiphilic azido linker with three PEG segments, to modify the dextran backbone before synthesizing the NPs. The distribution and orientation of the two linkers were investigated using a click-based colorimetric labeling method. Specifically, Cy5 molecules with dibenzocyclooctyne (DBCO) groups were clicked onto the naked NPs via click chemistry, and the overall density of azido group was determined by measuring the optical density (OD) value of Cy5 on the NPs. Higher OD indicates a higher number of azido

functional groups on the surface of naked NPs, whereas lower absorption suggests that more azido groups are oriented inside the NPs (Fig. 1b). The amphipathic linker with PEG segments orients more at the water–organic solvent interface, producing a higher density of azido groups on the NP surface compared to NPs without PEG segments (Fig. 1c). This linker was applied for further use. The available azido groups on the surface of NPs for antibody conjugation was 24.6 nmol per mg dextran NP as determined by UV-Vis spectroscopy analysis (Supplementary Fig. 8c, d).

Next, DBCO modified anti-CD3 and anti-CD28 antibodies were confirmed by nanodrop measurement, which showed the distinct absorption peak of DBCO from 300–320 nm, indicating that the DBCO linker has been conjugated to the antibody (Supplementary Fig. 9a, b). Furthermore, the DBCO modified antibodies were conjugated to the surface of the naked NPs through click chemistry. After the conjugation, the zeta potential decreased from $-8.5 \pm 0.25$ to $-12.5 \pm 0.87$ mV which is similar to the surface charge of Dynabeads™ (Fig. 1d).

To evaluate whether the conjugated anti-CD3 and anti-CD28 antibodies were successfully linked and retained their function for T cell activation, the T-Expand was co-cultured with primary T cells and analyzed for the expression of activation markers (CD25 and CD69) via flow cytometry. T cells were activated following formation of T cell clusters visible by microscopy (Supplementary Fig. 9c), with subsequent expression percentage of CD25 and CD69 similar to those induced by Dynabeads™ (Fig. 1e). In contrast, the naked NPs did not activate T cells, demonstrating the role of the conjugated antibodies on T-Expand.

To verify whether the surface stoichiometric ratio of anti-CD3 to anti-CD28 on T-Expand precisely mirrored the input feed, anti-CD3 was labeled with Cy5-NHS and anti-CD28 with Alexa Fluor 488-NHS, and both antibodies were simultaneously functionalized with DBCO-Sulfo-NHS (Supplementary Fig. 10a, b). In every case, the measured surface stoichiometry matched the input ratio, confirming feed-ratio control over ligand composition (Source data).

Next, to determine the absolute amount of antibody loading without potential competition between fluorophore and DBCO labeling, we employed a single bifunctional tag-anti-CD28 was derivatized with Cy5-DBCO-NHS (Supplementary Fig. 10c, d). Applying the calibration curve to the post-reaction supernatants allowed us to calculate the antibody mass bound per mg of dextran nanoparticle. As shown in Fig. 1f, T-Expand achieved an average loading of $18.2 \pm 3.1 \mu g$ ($4.5 \mu g$ anti-CD3, $13.6 \mu g$ anti-CD28) antibody per mg of naked NP. Compared to current T cell expansion scaffolds, our T-Expand achieves the highest antibody coupling density among reported nanoscale T cell expanders, even surpassing some micron-scale scaffolds (Supplementary Table 1). Additionally, cryogenic transmission electron microscopy (cryo-TEM) visualization (Fig. 1g) revealed that T-Expand exhibited the size of $173.1 \pm 37.27$ nm (Fig. 1h), while the NTA and dynamic light scattering (DLS) revealed size distribution at $204 \pm 45.8$ nm (Supplementary Fig. 11a) and $197 \pm 39.9$ nm (Supplementary Fig. 11b), respectively. This is because the physical diameter of the NP (as determined by cryo-TEM imaging) is typically smaller than its hydrodynamic diameter[18,19]. Overall, these data demonstrated that the click chemistry-mediated antibody conjugation method effectively anchors antibodies onto the naked NPs.

Next, we explored the presence of serum on T-Expand performance since several T-cell expansion protocols requires serum supplementation. The proteomics analysis revealed that the click chemistry-mediated antibody conjugation method also limits the protein corona formation on the surface of T-Expand (Supplementary Figs. 12, 13) and show no functional compromise for T cell activation (Supplementary Fig. 13h, i). Biocompatibility experiments indicated that neither naked dextran NPs nor T-Expand affect cell viability when co-cultured with human Jurkat T cells (Supplementary Fig. 14).

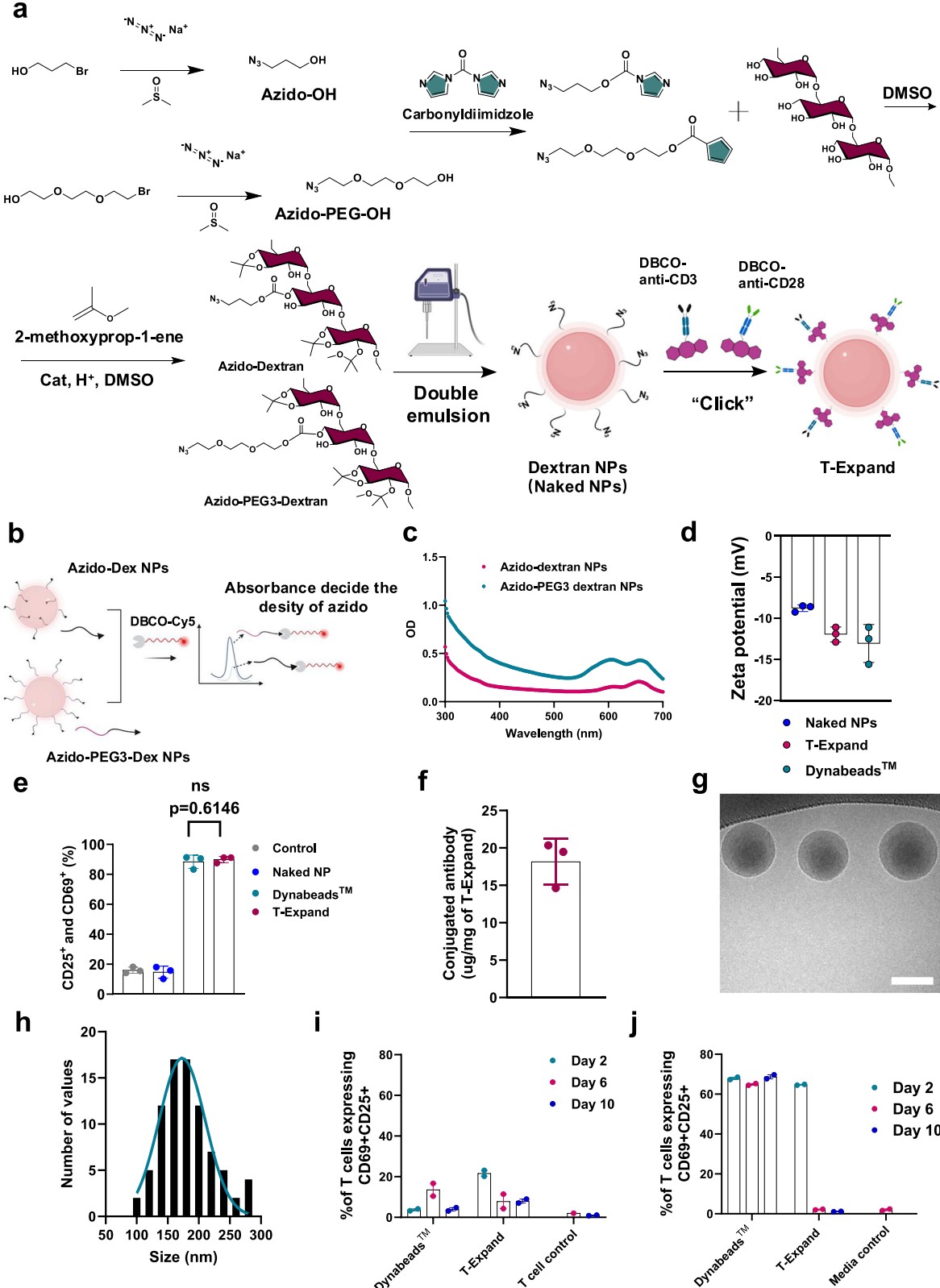

We further assessed the stability of T-Expand under physiologically relevant conditions. First, T cells were expanded with either T-Expand (20 μL, 0.75 mg/mL) or Dynabeads™ (75 μL/mL, bead-to-cell ratio 3:1), and culture supernatants were harvested at various time points. These supernatants were diluted 1:1 and applied to fresh, resting primary T cells. After a 20-hour incubation, T-cell activation was evaluated by co-expression of CD69 and CD25 (Fig. 1i). The stimulatory potency of the T-Expand supernatants declined progressively from day 6 onward. To eliminate any cell-mediated contributions, we next incubated T-Expand and Dynabeads™ separately in cell-free medium. These cell-free supernatants were then directly used to stimulate primary T cells. Under these conditions, T-Expand lost significant activity by day 6 (Fig. 1j), whereas Dynabeads™ maintained relatively stable stimulatory capacity.

**Fig. 1 | Design and characterization of anti-CD3 and anti-CD28 conjugated dextran-based T-Expand. a** Synthesis of azido modified dextran polymer, and the fabrication of T-Expand. **b**, UV-spectrum confirming that the azido linker is primarily orientated on the surface of naked NPs. **c** UV-spectrum showing azido decorated dextran NPs in solution clicked with DBCO-Cy5 to determine the density of azido linkers. **d** Zeta potential of naked NPs, T-Expand and Dynabeads™ (mean ± S.D., from $n = 3$ independent technical measurements). **e** Percent of T cells expressing the activation markers CD25 and CD69 after co-incubation with T-Expand (mean ± S.D., from $n = 3$ biologically independent donors), Statistical significance determined via unpaired $t$-test. **f** The antibody conjugation efficiency on the surface of naked NPs measured by microplate reader (mean ± S.D., from $n = 3$ independent technical measurements). **g** Cryo-TEM visualizing of T-Expand, scale bar: 100 nm. **h** The size distribution of T-Expand, as determined by Cryo-TEM and measured by software of image J with 3 independent figures. **i** the T cell activation profile in terms of % of T cells expressing CD69+ CD25+ after 20 hours of co-incubation with the supernatant obtained from the expansion cultures at different time point ($n = 2$; technical replicates from the same donor). **j** The T-Expand (20 μl, 0.75 mg/mL) and Dynabead™ (75 μl/mL) were incubated in culture medium at 37 °C and 5% $CO_2$ for 10 days and the media was removed at day 2, 6, and 10 timepoints of culture and was added to non-activated T cells at 1:1 dilution($n = 2$; technical replicates from the same donor). Data in (**i**, **j**) represented as mean ± SD, of biological replicates with each dot indicating one healthy donor sample. Significance was defined as $p < 0.05$, ns, not significant. Source data are provided as a Source Data file. **a**, **b** Were created with BioRender.com released under a Creative Commons Attribution (CC BY) 4.0 International license.

Additionally, T-Expand could be effectively removed by standard T-cell washing procedures due to the nanoscale of T-Expand, obviating the need for magnetic bead separation as required for Dynabeads™. Transmission electron microscopy of T cells harvested at various time points post–T-Expand activation further confirmed the absence of residual T-Expand both on the cell surface and in the post-wash supernatant (Supplementary Fig. 15).

Together, these data indicate that the active components of T-Expand gradually degrade or become depleted over time. Importantly, residual T-Expand that has not been fully removed does not activate T cells and can be readily cleared by simple centrifugation and washing, thereby underscoring its applicability for CAR T cell manufacturing.

## Investigation of the interaction of T-Expand with T cells

To study the interaction of T-Expand with T cells, we engineered Jurkat T cells to express a cathepsin L-mCherry fluorescent reporter. Cathepsin L is a lysosomal protein co-localizing with perforin and Granzyme B in T cell granules[20]. Using the Jurkat cathepsin L-mCherry reporter system, we investigated the ability of T-Expand to activate T cells by viewing the movement of cathepsin L-mCherry granules. Confocal imaging of engineered Jurkat cells in 3D and simulation shows the mCherry-Cathepsin L reporter has been successfully expressed in Jurkat cells (Fig. 2a). Colocalization imaging indicated that T-Expand effectively activated Jurkat cells, accompanied by the translocation of mCherry-Cathepsin L to the T-Expand contact site, ultimately resulting in colocalization. In contrast, the naked NPs did not directly interact with the Jurkat cells (Fig. 2b). Furthermore, we combined a 3D perspective simulation (Fig. 2c) and measured the distance between the T-Expand and the granules (Fig. 2d). The distance between T-Expand and the granules suggests colocalization, underscoring the specificity of T-Expand in activating Jurkat cells. The real time-3D imaging shows that the T-Expand forms a multicentric granules cluster that is different from the monocentric granule formed with Dynabeads™ (Fig. 2e, Supplementary Movie 1&2), indicating that T-Expand could activate Jurkat cells at multiple interfaces between T-Expand and Jurkat cells.

## Manufacturing of CAR T cells with T-Expand results in cell products with high proliferative capacity, cytotoxicity and persistence

We investigated two parameters known to influence T cell expansion: the ratio of conjugated anti-CD3:anti-CD28 antibodies on T-Expand and the dose of T-Expand. First, we investigated how the stoichiometric ratio of anti-CD3:anti-CD28 influences T cell activation. Optimization experiments revealed that a Dynabeads™:T-cell ratio of 3:1 yielded the most efficient CAR-T cell production (Supplementary Fig. 16a–h). Consequently, this ratio was selected as the benchmark condition for comparison with T-Expand. To avoid potential interference caused by variations in the concentration of anti-CD3 and anti-CD28, we ensured that the surface density of these factors was saturated in all experiments. Based on the T cell activation assay, the different stoichiometric ratios of anti-CD3:anti-CD28 antibodies did not influence significantly on the percentage cells positive for the activation markers CD25 and CD69. For all stoichiometric ratios, the percentage of these activation markers increased with higher concentrations of T-Expand, reaching saturation at 3.75 μg of T-Expand (0.75 mg/mL) per $0.5 \times 10^6$ T cells (Fig. 3a). At lower concentrations of T-Expand, we observed that T-Expand (anti-CD3:anti-CD28 ratio 1:3) demonstrated the highest fold expansion by day 14 (Supplementary Fig. 17a–c).

Then we followed the viability and fold expansion of the T cells over a 10-day cultivation period and compared T-Expand with gold standard of Dynabeads™ with a beads to cell of 3:1, a commonly-recommended dose[21–25], and TransAct™ as the standards. (Fig. 3b, c). T-Expand at the dose of 15 μg induced T cell expansion at a similar degree as Dynabeads™ by day 10 and superior to TransAct™ (Fig. 3c). The T-Expand-expanded T cells predominantly consisted of 35–50% effector memory T cells (TEM) and approximately 40% effector memory T cells re-expressing CD45RA (TEMRA), which mirrored the distribution observed in Dynabeads™-expanded T cells (Supplementary Fig. 17d). Notably, this condition yielded higher CD27+ CD28+ expression (Fig. 3d) and lower levels of PD1+ and LAG3+ on T cells (Fig. 3e). Consequently, the dose at 15 μg T-Expand /million T cells was selected for future use.

Next, the efficacy of T-Expand for engineering T cells with a CAR was evaluated. Activation with T-Expand prior to viral transduction resulted in a T cell transduction efficacy of ~50% CAR expressing T cells when infected at a multiplicity of infection (MOI) of 3 (MOI-3), which was similar to levels obtained with Dynabeads™. In contrast, activation with TransAct™ resulted in 20% transduction efficiency (Fig. 3f) and an overall lower yield of T cells (Supplementary Fig. 18a) after 10 days of expansion (Fig. 3g). Phenotypic analysis revealed that T-Expand CAR T cells had a significantly higher percentage of CD27+ CD28+ CAR T cells (Fig. 3h–j) and higher percentage of TEM (Fig. 3k and Supplementary Fig. 18b) with an increased CD8 + CAR T cell population compared to Dynabeads™ and TransAct™ (Supplementary Fig. 18c, d). For further functional analysis, we compared T-Expand against Dynabeads™ expanded CAR T cells.

To determine the proliferative potential of anti-CD19 CAR T cells after expansion with T-Expand, expanded and cryopreserved CAR T cells were stained with CellTrace Violet and stimulated with irradiated CD19+ Jeko-1 cells to assess their expansion potential upon target recognition (Fig. 4a). On day 6, flow cytometry analysis showed that ~85% of T-Expand expanded CD19 CAR T cells had undergone up to four rounds of population doublings, in comparison to only 20% of Dynabeads™-expanded CAR T cells (Fig. 4b). To determine the cytotoxic potential of CAR T cells expanded with T-Expand, we performed an incucyte based cytotoxic assay with co-cultivation of CAR T cells and Jeko-1 cells at effector:target ratio (E:T) 5:1; 1:1; 1:5 or 1:10. (Fig. 4c). CAR T cells expanded by T-Expand exhibited potent cytotoxicity against CD19+ Jeko-1 cells. Compared to CAR T cells expanded with Dynabeads™, CAR T cells expanded with T-Expand showed significant

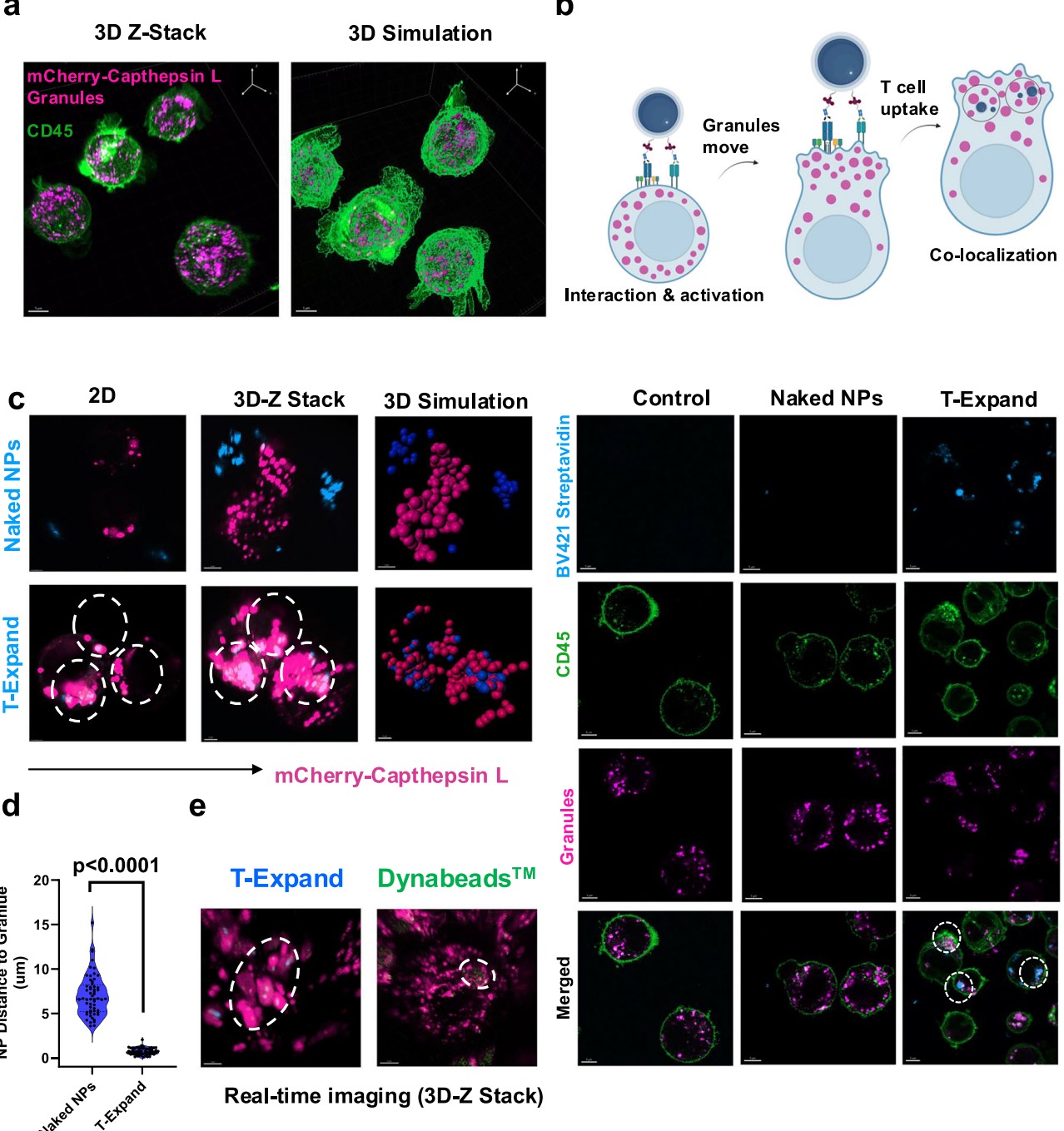

**Fig. 2 | The investigation of T-Expand interacting with Jurkat cells. a** mCherry labelled Cathepsin L is located in Jurkat cell granules as shown by confocal imaging, scale bar: 5 μm, representative confocal images from 3 independent experiments performed with similar imaging results. **b** Engineered Jurkats were incubated with T-Expand or naked NPs both loaded with BV421-streptavidin for 24 h and then imaged by Airyscan confocal imaging (2D-view), scale bar: 5 μm. White circle indicates the location of naked NPs or T-Expand. **c** Specific activation test, Airyscan imaging (3D view) and Imaris 3D simulation on how T-Expand interacting with engineered Jurkats, scale bar 2 μm. **d,** The distance of naked NPs or T-Expand to mCherry-Cathepsin L under Imaris 3D simulation, (number of granules $n = 130$), two tailed unpaired $t$ test, and significance was defined as $p < 0.05$. **e** The real-time 3D confocal imaging indicates the granules cluster formation between Dynabeads™ and T-Expand. White circle indicates the location of T-Expand or Dynabeads™, scale bar 3 μm. **b** Was created with BioRender.com released under a Creative Commons Attribution (CC BY) 4.0 International license. Significance was defined as $p < 0.05$, ns, not significant.

tumor cell-killing efficacy across all E:T ratios. To further investigate the serial cytotoxicity and phenotypic effect on CAR T cells after serial exposure to target cells, we performed a re-challenge killing assay, by continuously exposing remaining CAR T cells with CD19 target cells every 2-3 days. Based on the E:T ratio at 5:1, T-Expand-expanded CAR T cells exhibited significantly higher killing capacity even after three rounds of re-challenge (Fig. 4d, e, and Supplementary Fig 19).

To further characterize CAR T cells with enhanced cytotoxic capacity after repetitive stimulation, remaining cells from the third re-challenge were harvested and analyzed by flow cytometry (Supplementary Fig 19a). T cell expanded with Dynabeads™ primarily had a terminally differentiated phenotype whereas CAR T cells expanded with T-Expand were primarily effector memory (Fig. 4 f, Supplementary Fig 19b). Additionally, the percentage of the CAR T cells

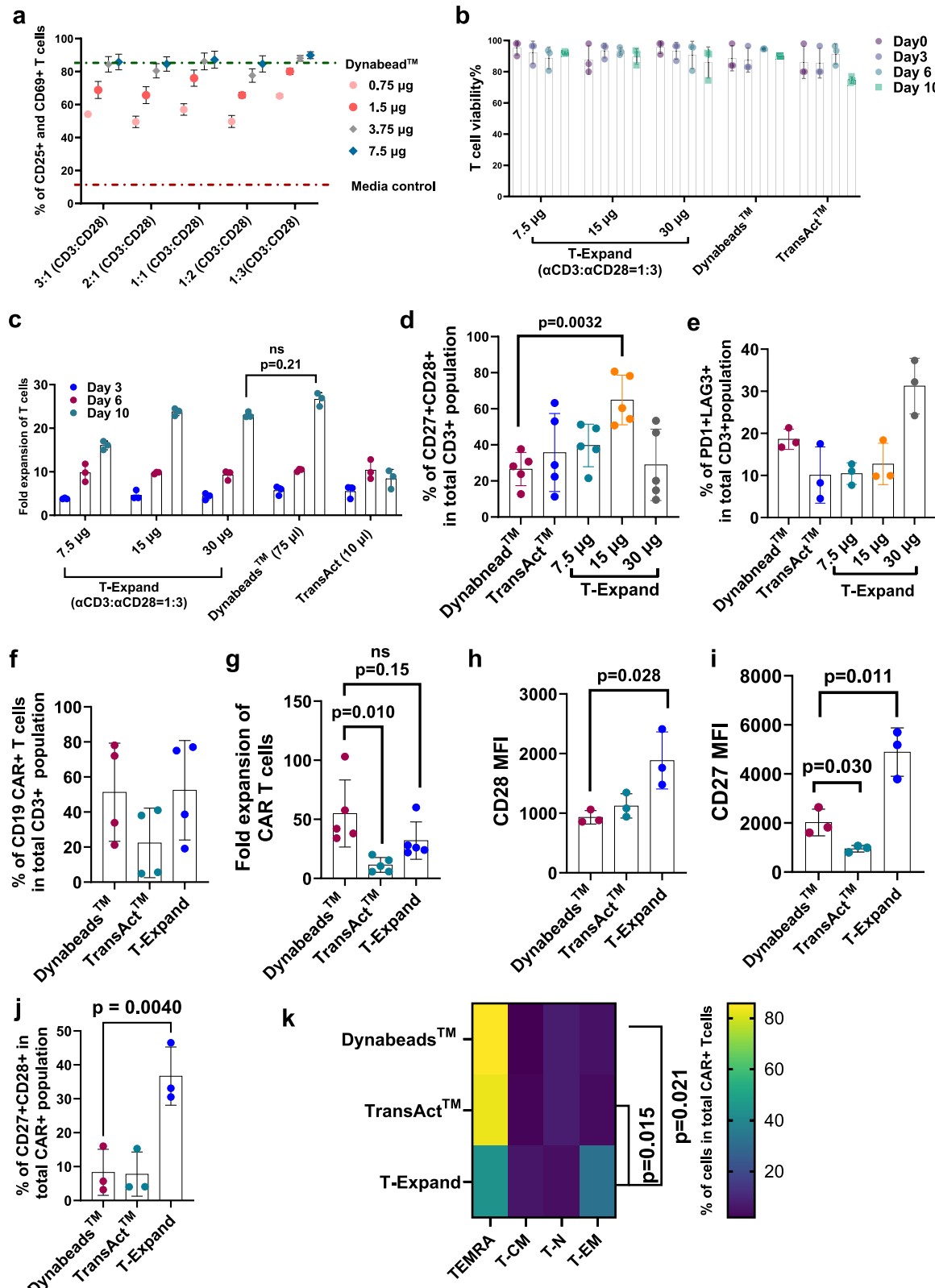

expressing exhaustion markers such as PD1, LAG3, TIM3, and TIGIT were all significantly lower in T-Expand compared with Dynabead™ expanded CAR T cells (Fig. 4 g, Supplementary Fig 19c). Next, we investigated the cytotoxicity of CAR T cells expanded using T-Expand under low E:T ratios. Similarly, CAR T cells expanded using T-Expand demonstrated significantly higher cytotoxicity, even after three rounds of re-challenge (Fig. 4h). Flow cytometry analysis revealed that

under co-culture conditions with E:T ratios of 1:1 and 5:1, the T-Expand group exhibited T cell persistence rates of approximately 70% and 99%, respectively, whereas the Dynabeads™ group showed only 0.2% and 45%. (Fig. 4i, Supplementary Fig 19d). This suggests that the superior cytotoxic capacity of T-Expand CAR T cells is likely attributed to a combination of enhanced proliferative capacity, effector memory phenotype and lower exhaustive profile.

**Fig. 3 | CAR T cell manufacturing with T-Expand results in T cell products with a favorable phenotype for ACT. a** T cell activation marker (CD25 and CD69) expression percentage as determined by flow cytometry following incubated with T-Expand with different ratio of anti-CD3: anti-CD28 for 24 h (number of donors: n = 3, presented as mean ± SD). **b** A graph showing the viability of T cells on the different time points of expansion with different dose of T-Expand, Dynabeads™, or TransAct™ (n = 3; biological replicates); **c** Graph showing the fold expansion of isolated T cells from peripheral blood expanded using T-Expand, Dynabeads™ or TransAct™ (n = 3; biological replicates). **d** The expression CD27+ and CD28 on T cells expanded by T-Expand (conjugated anti-CD3: anti-CD28 = 1:3) at different dose (7.5 μg, 15 μg, 30 μg per million Pan T cells), Dynabeads™, and TransAct™ (n = 5; biological replicates). **e** The expression of the exhaustion markers of PD1 + LAG3+co-expression on T cells expanded by T-Expand (anti-CD3: anti-CD28 ratio 1:3) at different doses (7.5 μg, 15 μg, 30 μg), Dynabeads™, and TransAct™ (n = 3; biological replicates). **f** Percentage of CD19 CAR T cells in cultures after after lentiviral transduction and 10 days of expansion (n = 4; biological replicates). **g** Fold expansion of CAR T cells 10 days expansion with T-Expand, Dynabeads™ or TransAct™ (n = 5); (**h–j**) Bargraph showing mean fluorescent intensity (MFI) of (**h**), BUV737-CD28, and (**i**), BV605-CD27on CAR T cells after expansion. **j** Percentage of CD27 + CD28 + CAR T cells after 10 days expansion with T-Expand, Dynabeads™ or TransAact™ (n = 3; biological replicates). **k** Heat map of phenotype analysis (naïve T cell: TN; TCM, TEM, TEMRA) of expanded CAR T cell during 10 days of expansion with T-Expand, Dynabeads™ or TransAct™ (n = 3; biological replicates). Data panel in b presented as mean ± SD, of biological replicates with each dot indicating one healthy donor sample (n = 3; biological replicates). Data in (**c–i**), presented as mean ± SD, of biological replicates with each dot indicating one healthy donor sample. Statistical significance is determined via unpaired t test. Data (**j**) presented as mean ± SD, of biological replicates with each dot indicating one healthy donor sample (n = 3). Statistical significance is determined via paired t test. Data panel k presented as mean ± SD, of biological replicates with each dot indicating one healthy donor sample (n = 3). Statistical significance determined via two-way ANOVA using Dunnett's multiple comparisons test, the statistical significance is mentioned for T-EM phenotype between the expansion conditions. Significance was defined as p < 0.05, ns, not significant. Source data are provided as a Source Data file.

## T-Expand expanded CAR T cells show superior in vivo tumor killing efficacy

Subsequently, we tested the in vivo antitumor efficacy of the CD19 CAR T cells expanded with T-Expand in NXG mice injected with the lymphoma cell line Jeko-1 engineered to express luciferase (luc-mCherry + ) for monitoring tumor burden using bioluminescence (BLI). Initially we tested a dose of $1 \times 10^6$ CAR T cells expanded with either T-Expand or Dynabeads™ (Fig. 5a). The tumor growth measured by BLI shows that, the treatment with T-Expand expanded CAR T cells lead to complete tumor elimination by day 7 without signs of tumor relapse at study termination on day 21. However, the Dynabead™ expanded CAR T cell treated group showed insufficient tumor clearance on day 7, further leading to the gradual increase in tumor burden until day 21. (Fig. 5b, c and Supplementary Fig. 20). At study termination, mice were euthanized and spleen and bone marrow were collected for detection of CAR T cells. The results show that the percentage of the CAR T cells detected both in bone marrow and spleen is similar between the T-Expand and the Dynabeads™ group (Fig. 5d). Interestingly, mice treated with T-Expand expanded CAR T cells had a significantly higher proportion of CD4+ T cells in the spleen (Fig. 5e) and higher levels of CD137 (4-1BB) in bone marrow compared to Dynabeads™ expanded CAR T cells (Fig. 5f). CD137 expression on T cells serves as a marker for antigen specific activation often correlated with enhanced anti-tumor efficacy. CD137 receptor interaction with 4-1BB ligand also plays a significant role in enhancing the survival and proliferation of these T cells[26]. Next, to further elucidate differences between CAR T cells expanded with T-Expand or Dynabeads™, mice were injected with $1 \times 10^6$ luc-mCherry+ Jeko-1 cells and treated on day 7 with $0.25 \times 10^6$, $0.5 \times 10^6$ or $1 \times 10^6$ CAR T cells expanded with either Dynabeads™ or T-Expand (Fig. 5g). On day 21, blood was collected for further analysis. A higher percentage of circulating T cells in mice receiving CAR T cells expanded with T-Expand (Supplementary Fig. 21a). The mice treated with T-Expand expanded CAR T cells at lowest doses ($0.25 \times 10^6$ and $0.5 \times 10^6$) showed some tumor control and the high dose ($1 \times 10^6$) completely eradicated the tumors (Fig. 5h). In contrary, Dynabeads™ expanded CAR T cells at all doses failed to show any significant tumor control with the majority of mice developing systemic hematological tumors until day 33 (Fig. 5i, Supplementary Fig. 21b).

## RNA sequencing analysis of expanded CAR T cells

To investigate the transcriptomic profile of expanded CAR-T cells, we collected CAR T cells, generated from four donors, on Day 10 of expansion using either T-Expand or Dynabeads™ for bulk RNA sequencing from four healthy donors. The principal component analysis (PCA) revealed that the overall gene profiles of T-Expand expanded CAR T cells markedly differed from those of Dynabeads™ expanded CAR T cells (Supplementary Fig. 22a, b). T-Expand expanded CAR T cells exhibited high expression of genes associated with a putative transitional state between cytotoxic effector and differentiation phenotypes (Fig. 6a). These transitional phenotypes were identified by the co-expression of multiple cytotoxic genes, including *LTB* and *GZMA*[27].

The analysis showed that the T-Expand expanded CAR T cells also showed upregulation of *Xcl1* and *Xcl2* genes that are involved in dendritic cell recruitment and activation. The stem-like T cells express high levels of *Xcl1* that promotes the infiltration of migratory cDC1s that are associated with improving CD8 T cell anti-tumor activity through cross-primin[28,29]. Among the selected gene set (Fig. 6b), CAR T cells expanded with T-Expand showed increased expression of genes associated with anti-tumor cytokines and effector molecules (*IFNg, GZMA*), as well as T cell infiltration and activation (*CXCL9, CXCL10, CCL5*), compared with Dynabeads™ expanded CAR T cells[30–32] (Supplementary Fig 22 b).

To further elucidate the global transcriptomic landscape related to T-cell functionality in T-Expand expanded CD19 CAR T cells, only significantly expressed genes (p.adj < 0.001) were selected, and the top 10 gene sets were characterized via gene set enrichment analysis (GSEA) (Fig. 6c). The results revealed a significant enrichment of gene sets related to immune responses, inflammatory regulation, and signal transduction in T-Expand expanded CAR T cells. In contrast, Dynabeads™ expanded CAR T cells did not show notable enrichment of these gene sets, indicating substantial differences in gene expression regulation between these two expansion strategies.

## Discussion

In this study, we successfully developed T-Expand, a unique nanoplatform designed for the efficient expansion of T cells. Compared to existing artificial antigen-presenting nanoparticle carriers (Supplementary Table 1), we employed azido-modified dextran polymers to install a high density of azido groups along the side chains for downstream antibody conjugation. In contrast, current polymer platforms, such as poly(lactic-co-glycolic acid and poly(ethylene glycol)-block-poly(d,l-lactide), typically localize azido groups at a single terminus of the polymer chain, yielding a low surface functional-group density after NP assembly. Moreover, our azido-functionalized dextran NP requires no additional surface modification to undergo click chemistry with DBCO-anti-CD3 and DBCO-anti-CD28, thereby providing an efficient, convenient, and controllable antibody-conjugation system[17,33–35]. In addition, these azido-modified dextran nanoparticles remain stable in PBS buffer without loss of azido-group reactivity, offering a ready-to-use antibody-clickable nanoplatform. We then optimized the

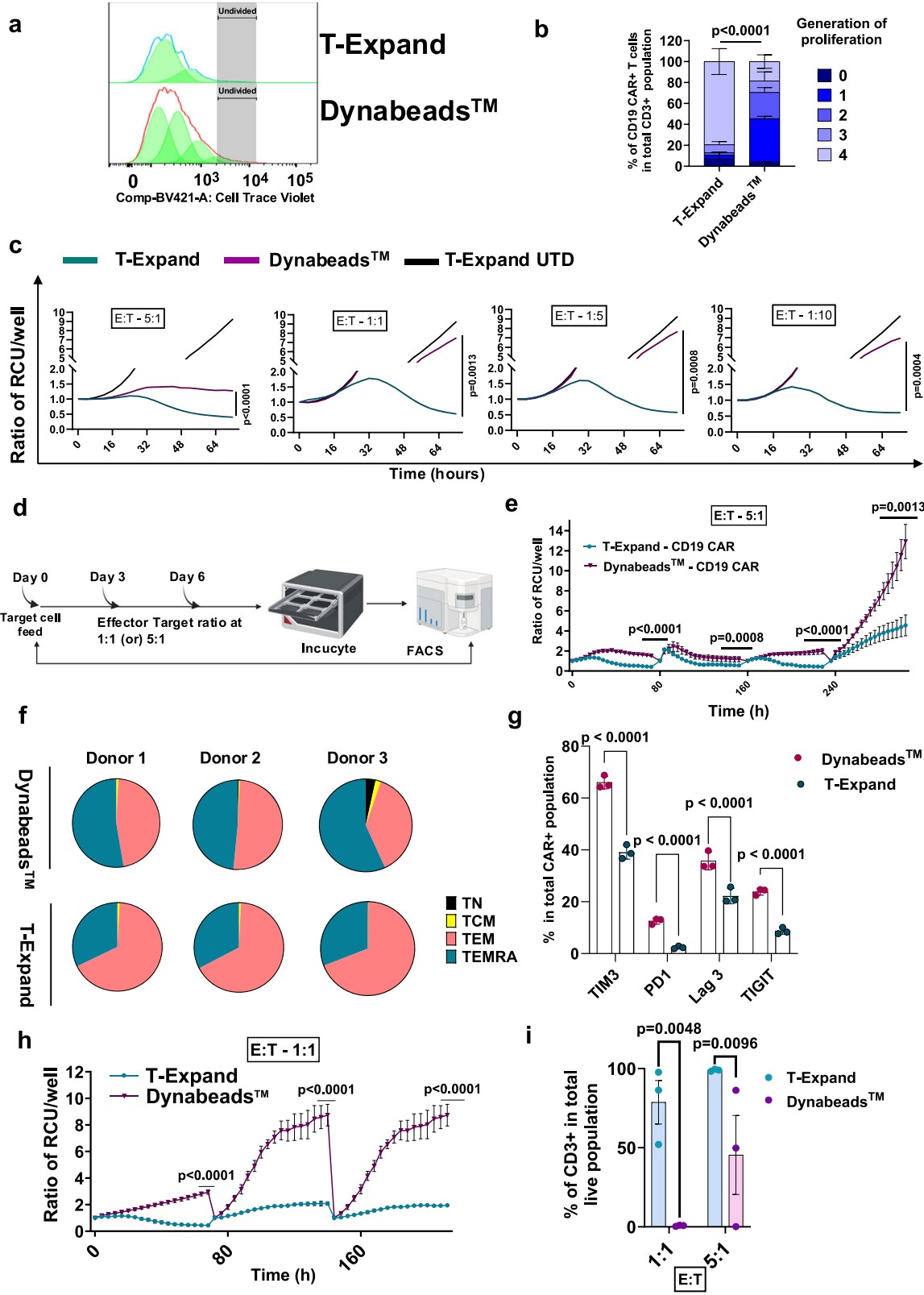

stoichiometric ratio of anti-CD3 and anti-CD28 antibodies, achieving both robust T cell yields and favorable phenotypic characteristics of the expanded T cell products. Furthermore, the T-Expand's great biocompatibility and nanoscale advantages eliminate the need for additional removal steps following T-cell expansion.

We employed genetically engineered Jurkat cells expressing mCherry-Cathepsin L to visualize how T-Expand induces Jurkat cell

activation. Leveraging the multifunctionality of T-Expand, we loaded BV421-streptavidin into T-Expand to track its interactions with T cells. Once naïve T lymphocytes are activated, CD4+ T cells, similar to CD8+ T cells, polarize labeled lysosomal granules toward their target cells[36]. Based on this observation, when T-Expand interacts with Jurkat cells, the uniformly distributed mCherry-labeled Cathepsin L granules co-localize and form prominent clusters around T-Expand. Comparative

**Fig. 4 | T-Expand enhanced the proliferation and cytotoxic capacity of anti-CD19 CAR T cells. a** Representative histograms showing the proliferation peaks on Day 6 of restimulation. Proliferation is determined by flow cytometry analysis (proliferation index in Flowjo by measuring the dilution of cell trace violet. **b** graph showing the percentage of CAR T cells in different generations of proliferation quantified based on the dilution of cell trace violet and proliferation index analysis. ($n = 3$; biological replicates)**c**, Cytotoxicity of CAR T cells in co-culture with Jeko-1 cells at an E:T ratio 5:1; 1:1; 1:5 or 1:10; tracked over 72 h in the Incucyte, represented by ratio of relative confluence units (RCU) per well normalized to 0 h ($n = 3$; biological replicates). **d** Schematic illustration of the incucyte based cytotoxicity re-challenge assay. **e** Cytotoxicity of CAR T cells in co-culture with Jeko-1 cells at an E:T ratio of 5:1, tracked over 240 h in the Incucyte with re-challenge every 48 h, represented by ratio of relative confluence units (RCU) per well normalized to 0 h ($n = 3$; biological replicates). **f** Pie chart showing the percentage of CAR T cells with a CD45RA+ CCR7- terminally differentiated phenotype and CD45RA-CCR7- effector memory phenotype. **g** Graph showing the percentage of CAR T cells expressing PD-1; Tim3; LAG3 and TIGIT surface markers after 3 rounds of rechallenge with the Jeko-1 target cells ($n = 3$; biological replicates). **h** Cytotoxicity of CAR T cells in co-culture with Jeko-1 cells at an E:T ratio of 1:1, tracked over 200 h in the Incucyte with re-challenge every 48 hours ($n = 3$; biological replicates). **i** Bar graph showing the percentage of CD3+ T cells in the total live cell population at E:T ratio of 1:1 and 5:1 after three rounds of re-challenge ($n = 3$; biological replicates). Data panel in b represent results from three biological replicates presented as mean with ±SD, with each dot indicating one healthy donor sample and statistical significance determined via two-way Anova with Turkey's multiple comparison test, the statistical significance mentioned in the figure corresponds to the 4th generation of proliferation between the two expansion conditions. Data in panels c, e, h represent results from three biological replicates presented as mean ± SD, with each dot indicating one healthy donor sample statistical significance determined by paired *t* test. Data in panels g and i represent results from one experiment, three biological replicates presented as mean ± SD, with each dot indicating one healthy donor sample, statistical significance determined by unpaired t test. Significance was defined as p < 0.05, ns, not significant. Source data are provided as a Source Data file. **d** Was created with BioRender.com released under a Creative Commons Attribution (CC BY) 4.0 International license.

studies further showed that, relative to Dynabeads™-activated cells, Jurkat cells activated by T-Expand exhibit significantly enhanced multicentric granule trafficking and aggregation. These findings suggest that T-Expand can activate Jurkat cells at multiple sites and effectively induce granule polarization toward the T-Expand contact regions.

Activation with T-Expand successfully lead to virally transduced CAR expression in approximately 50% of T cells, achieving similar results as Dynabeads™ and outperformed TransAct™. Our results demonstrate that T-Expand-expanded CAR T cells exhibit a less differentiated phenotype, featuring a high proportion of CD27+ CD28+ effector memory T cells, compared to those generated using conventional platforms such as Dynabeads™ and TransAct™, which predominantly produced TEMRA cells. This indicates that T-Expand facilitates the generation of CAR T cells with a potentially more favorable phenotype for therapeutic efficacy.

Mechanistically, the divergent functional outcomes of T-Expand and Dynabeads™ appear to be driven by differences in stimulus intensity and kinetics. T-Expand presents a higher surface density of anti-CD3/anti-CD28, delivering more activation signals and enabling the rapid formation of multiple immunological synapses. This brief, high-density "burst" is sufficient to initiate T-cell activation and proliferation without prolonged high-strength engagement. By contrast, we observed that Dynabeads™ remain tightly connected to T-cell membranes throughout the expansion period, providing continuous strong signaling that while promoting early proliferation, may also predispose cells to activation-induced exhaustion. In addition, the size and biocompatibility of T-expand facilitates receptor-mediated internalization, which aligns with the natural T cell behavior upon TCR activation. We hypothesize that the internalization capacity contributes to the observed phenotype.

Consistent with this model, our stimulation-stability assays show that, under cell-free conditions, T-Expand's stimulatory activity declines by day 6, indicating natural dissipation/degradation that limits prolonged overstimulation, whereas Dynabeads™ maintain relatively constant potency. In line with this, CAR-T cells expanded with Dynabeads™ exhibit higher exhaustion-marker expression in a three-round in vitro tumor rechallenge assay, whereas those expanded with T-Expand retain a less-exhausted CD27+CD28+ phenotype and superior durability under repeated challenge.

In cytotoxicity assays, CAR T cells expanded by T-Expand demonstrated robust anti-tumor activity. The CD19 CAR T cells maintained high viability and consistent killing activity after three rounds of re-challenge with CD19+ Jeko-1 target cells. Phenotypic analysis post-rechallenge of the CAR T cells revealed a significantly increased proportion of effector memory T cells and reduced expression of exhaustion markers, indicating that CAR T cells possess enhanced durability and survival potential leading to more efficient anti-tumor responses. This optimal cellular profile is likely key to the superior functionality and persistence of T-Expand expanded CAR T cells as observed in our in vivo studies. Based on the lymphoma model, a dose of $0.25 \times 10^6$ CAR T cells expanded with T-Expand achieved robust anti-tumor efficacy, even outperforming the high-dose treatment group of $1 \times 10^6$ CAR T cells produced with Dynabeads™. This distinct tumor-killing efficacy in vivo aligns with our in vitro cytotoxicity and repetitive restimulation assays. CAR T cell products with a younger effector memory phenotype (CD27+ and CD28+) and heightened proliferation and cytotoxic capacity have demonstrated improved efficacy in clinical applications[37–41].

Bulk RNA sequencing of CAR T cells expanded using the T-Expand platform showed significant upregulation of key effector molecules (GZMA, EOMES, IFN-γ, TNFRSF9/CD137), aligning with their sustained cytotoxicity and robust rechallenge capacity observed in vitro and in vivo. These cells also showed elevated expression of Th2 (IL4, IL5, IL10, IL13), Th9 (IL9), and Th17 (IL17A, IL17F, IL22) cytokine genes. Recent studies suggest that Th2 CAR T cell products provide superior expansion and antitumor efficacy, particularly after leukemia relapse, and that enhancing Th2 functionality can partially restore antitumor effects[21,42,43]. T-Expand expanded CAR T cell treated mice showed significant increase in the proportion of CD4+ T cells in the spleens. Notably, T-Expand−expanded CAR T cells also showed high expression of multiple chemokine receptor genes (CCR1, CCR2, CCR3, CCR5, CCR6, CCR10) combined with the high expression of chemokines (CCL1, CCL3, CCL4, CCL20), which may facilitate the recruitment of other immune cells, further amplifying and modulating local immune responses as well as enhancing CAR-T cell infiltration into the tumor microenvironment[31]. However, we could not confirm these compartments of the anti-tumor activity due to the immunodeficient nature of our NSG mouse model. Overall, bulk RNA sequencing demonstrates significant gene expression differences between CAR T cells expanded using the T-Expand and those using Dynabeads™. These molecular features provide potential mechanistic support for the superior cytotoxicity and sustained anti-tumor effects observed with T-Expand expanded CAR T cells.

There is marked upregulation of pro-inflammatory cytokine transcripts (IFN-γ, IL-6, IL-17) in T-Expand−expanded CAR T cells relative to Dynabeads™. While such signatures are consistent with heightened effector programming, they overlap with pathways implicated in clinical cytokine release syndrome (CRS) and immune-effector cell−associated neurotoxicity syndrome (ICANS)[44]. Importantly, bulk transcriptomic elevation does not equate to systemic cytokine release, and in patients the surge in IL-6 during CRS is

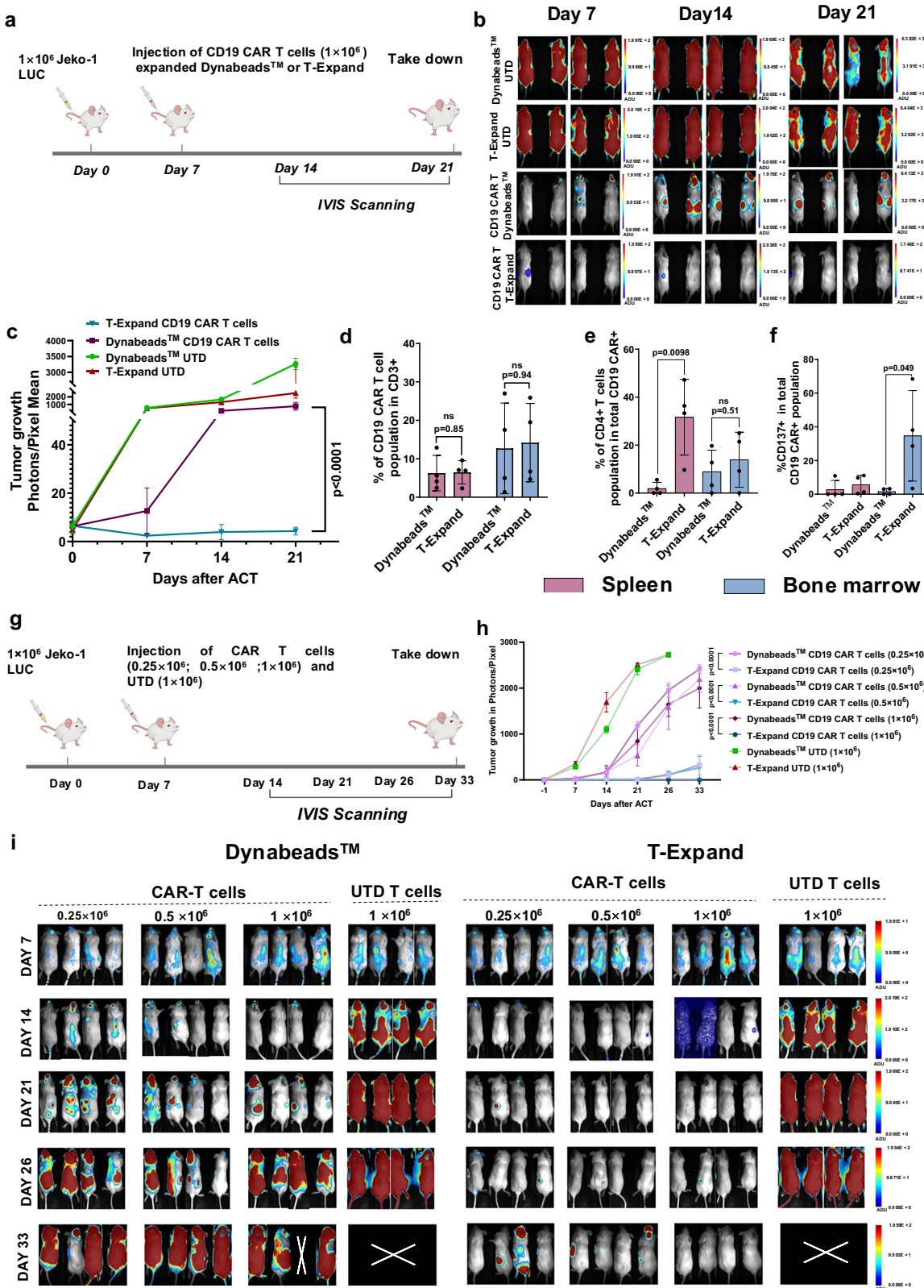

predominantly mediated by monocyte/macrophage activation rather than by T cells themselves[45,46].

In our NSG mouse xenograft experiments, we did not observe any adverse reactions, including distress, weight loss, behavioral changes, during the treatment period, despite the potent antitumor activity of T-Expand–expanded CAR T cells. Considering that NSG mice are not an appropriate system to model CRS and ICANS, this should be evaluated

in future studies using immune-competent and humanized animal models in which myeloid-driven cytokine biology is preserved.

In summary, T-Expand offers properties that are beneficial to T cells in the context of adoptive cell therapy. T-Expand not only efficiently produces a clinically relevant proportion of CAR T cells, but also creates a more beneficial T cell subset that is associated with enhanced proliferation, persistence, and anti-tumor efficacy by

**Fig. 5 | T-expand expanded CAR T display superior antitumor activity in a xenograft model of B-cell lymphoma. a** Schematic of the in vivo study setup. NXG mice were injected with $1 \times 10^6$ Jeko-1 lymphoma cells expressing mCherry and luciferase. Tumor volumes were measured on day 7 by bioluminescent imaging (BLI), and mice were randomized by tumor burden, receiving either T-Expand or Dynabeads™ expanded CAR T cells at indicated dosing or untransduced (UTD) T cells at the highest CAR T cell dose equivalent. Bioluminescence from luciferase-expressing tumor cells was tracked using IVIS imaging every 7 days. **b** Representative IVIS overlay images showing bioluminescent signals for each treatment group. **c** Bioluminescence intensity, reported as photons/pixel, tracked over time for each treatment group ($n = 4$). The mice were taken down on day 21 after ACT and the spleen and bone marrow were analyzed using flow cytometry. **d** Graph showing percentage of CD19 CAR T cells in the CD3+ T cell population in spleen and bone marrow ($n = 4$). **e** Graph showing percentage of CD4+ T cells in the CAR+ T cell population in spleen and bone marrow ($n = 4$). **f** Graph showing percentage of CD137+ T cells in the CAR+ T cell population in spleen and bone marrow ($n = 4$). **g** NXG mice were injected with $1 \times 10^6$ Jeko-1 lymphoma cells expressing

mCherry and luciferase. Tumor volumes were measured on day 7, and mice were randomized by tumor burden, receiving either CAR T cells at various doses or UTD T cells at the highest CAR T cell dose equivalent. Bioluminescence from luciferase-expressing tumor cells was tracked using IVIS imaging every 7–10 days. **h** Bioluminescence intensity, reported as photons/pixel, tracked over time for each treatment group with each line representing individual mice ($n = 4$). **i** Representative IVIS overlay images showing bioluminescent signals for each treatment group ($n = 4$). Data in (**d**–**f**) represents results from mice treated with CAR T cells, presented as mean ± SD with every dot or line representing a mouse. Statistical significance determined via unpaired *t*-test. Data in (**c**, **h**) represent results as mean ± SD with four mice in each group. Statistical significance determined via two-way anova with Šídák's multiple comparisons test, the mentioned statistics refers to the last time point (take down day) of each group. Significance was defined as $p < 0.05$, ns, not significant. Source data are provided as a Source Data file. **a**, **g** Were created with BioRender.com released under a Creative Commons Attribution (CC BY) 4.0 International license.

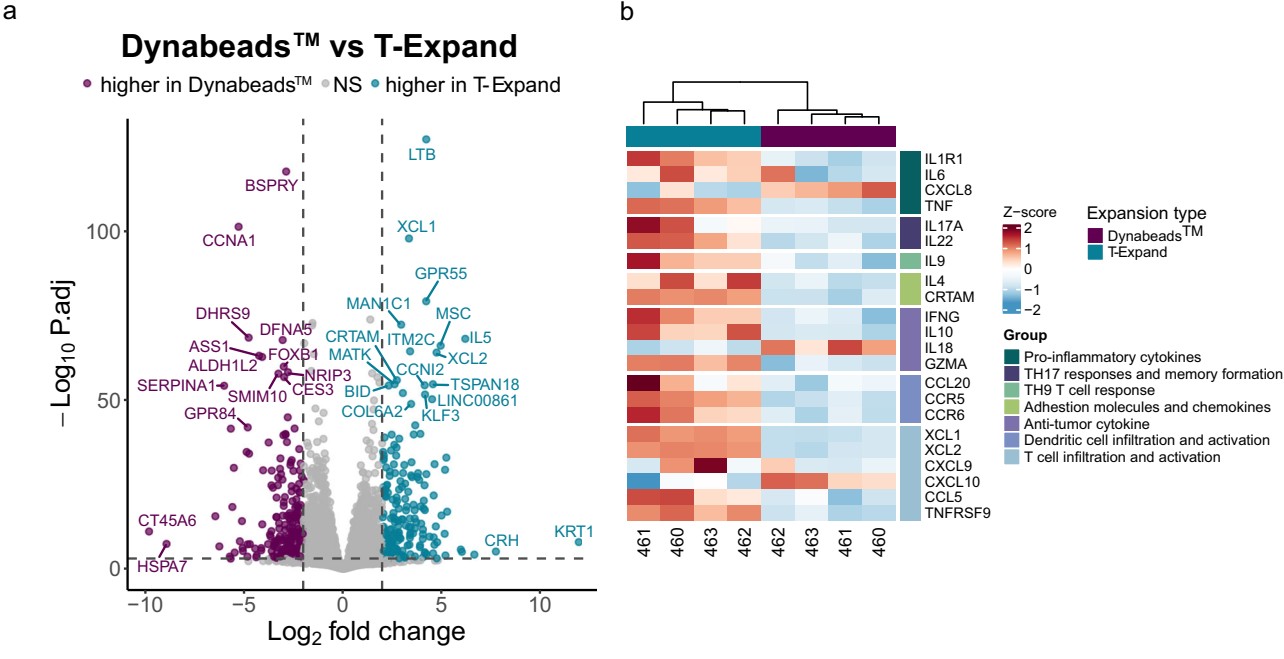

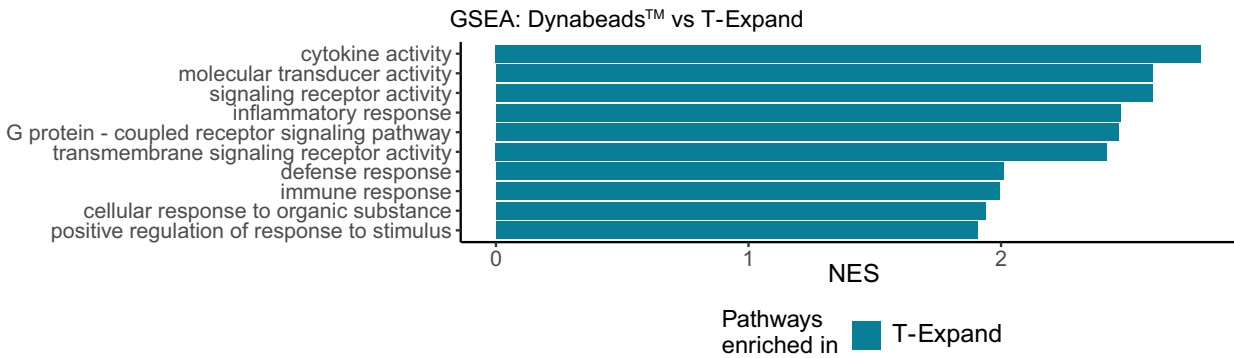

**Fig. 6 | Differential gene expression in CAR T cells expanded with either T-Expand or Dynabeads™, highlighting key changes between the two expansion methods. a** Volcano plot of differentially expressed genes between T-Expand and Dynabeads™ (identified using DESeq2 with the Wald test for statistical significance). **b** Heatmap showing selected genes grouped based on known functions. **c** GSEA was performed on bulk RNA-sequencing data. Normalized Enrichment

Score (NES) plot of the top 10 gene sets found in T-Expand. To keep only highly significant gene sets, we used only gene sets with >30 genes and with *p* values < 0.001. Gene-set enrichment was assessed using a pre-ranked GSEA with adaptive multilevel MonteCarlo permutation, and *p* values were adjusted using the Benjamini-Hochberg method.

modulating cell phenotype and reducing exhaustion. Thus, T-Expand offers a unique approach to ensuring the production of high-quality CAR T cells and underscores the potential of T-Expand as a promising alternative to traditional activation strategies.

# Methods

## Chemicals

Sodium azide (for synthesis), 3-Bromo-1-propanol (97%), dimethylformamide (ACS reagent, ≥99.8%), ethyl acetate (ACS reagent, ≥99.5%), magnesium sulfate (anhydrous, ReagentPlus®, ≥99.5%), Carbonyldiimidazole (CDI, ≥97.0% (T), for peptide synthesis), dextran (from Leuconostoc mesenteroides, average mol wt 9000–11,000), anhydrous dimethyl sulfoxide (DMSO), pyridinium p-toluenesulfonate (98%), 2-methoxypropene (97%), spermine (≥99.0% (GC)), Dimethyl sulfoxide-d6, poly(vinyl alcohol) (Mw 9000–10,000, 80% hydrolyzed) were purchased from Sigma-Aldrich.

## Preparation of dextran NPs (naked NPs)

Acetalated azido modified Dextran (10 KDa, 20 mg) was dissolved in cold dichloromethane (DCM, 500 μL). The PBS (100 μL, from DTU life science) was added to undergo a sonication for 20 s every 10 s on ice using a probe sonicator 9 (Qsonica L.L.C, USA) flat tip, an output setting power of 5. Then the poly(vinyl alcohol) (PVA, Mw 9000–10,000 g/mol, 80% hydrolyzed) (1 mL, 3% w/w in PBS) was added into the primary W/O solution and further sonicated for an additional 20 s on ice using the same settings. The resulting W/O/W emulsion was immediately transferred to the PVA solution (5 mL, 0.3% w/w in PBS) and stirred for 2 h at room temperature for evaporating the DCM solvent. The NPs were obtained by centrifugation (10,600 × g, 10 min, Heraeus fresco 21, Thermo Fisher Scientific) and washed with PBS 3 times.

## Formation of DBCO functionalized antibody and T-Expand

First, the anti-CD3 (clone OKT-3, BioXCell) and anti-CD28(clone 9.3, BioXCell) antibodies were dissolved in PBS buffer (0.5 mg/ml, 200 μl), and the DBCO-Sulfo-NHS (CCT-A124, Vector Laboratories) was dissolved in DMSO to final concentration at 10 mM. Next, 3.32 μL of the above DBCO-Sulfo-NHS Ester was added to each antibody solution and incubated at room temperature for 30 min. Then, the reaction was quenched with 18 μl of Tris-buffer (1 M, pH 8, Thermo Fisher Scientific) for 15 min. Finally, any unbound DBCO-Sulfo-NHS Ester was removed using Zeba spin desalting columns (7 kDa or 40 kDa MWCO, Thermo Fisher Scientific) and final dissolved in 200 μl PBS.

Then, the above DBCO-functionalized antibodies (DBCO-anti-CD3 and DBCO-anti-CD28) were conjugated onto the surface of azido modified dextran NPs via click chemistry. The DBCO-anti-CD3 (50 μl, 0.38 mg/ml) and DBCO-anti-CD28 (150 μl, 0.38 mg/ml) was mixed with naked NPs (100 μl, 3 mg/ml) at final volume of 600 μl PBS according to the ratio of 1:3. Similarly, naked NPs were conjugated with anti-CD3 and anti-CD28 at volume ratios of 3:1 (150 μL:50 μL in PBS) and 1:1 (100 μL:100 μL in PBS) to generate T-Expand formulations with different surface antibody ratios. After the incubation in room temperature for 2 h, the T-Expand was obtained by centrifugation (10.6 × g, 10 min) and wished with PBS 2 times, followed stored in 4-degree fridge for further using.

## Nanoparticle characterization

The number and size of the NPs was measured by Nanoparticle Tracking Analysis (NTA, NANOSIGHT, Malvern Instruments). The aliquot of NPs solution was diluted in MiliQ water pre-filtered by 20 nm filter at the same numbers used for expansion. The setup for NTA test was shown as follows (Camera Type: sCMOS Laser Type: Green; Camera Level: 10; Slider Shutter: 696; Slider Gain: 55; FPS 25.0; Number of Frames: 1498; Temperature: 23.9 °C; Viscosity: (Water) 0.910 −0.911 cP; Syringe Pump Speed: 100).

## Szie and ζ-Potential measurement of T-Expand

Hydrodynamic diameter of T-Expand were determined by Dynamic Light Scattering (DLS) (Malvern DLS Zetasizer, Malvern Panalytical) at 25 °C. T-Expand (10 μL, 0.75 mg/ml) were dispersed in MiliQ water (2 ml) with gently sonicated water bath for 1 min and loaded into disposable cuvettes. For each sample, three independent measurements were recorded and reported as mean ± SD ($n = 3$). Solvent viscosity and refractive index were set to those of water at the measurement temperature. Zeta potential of naked dextran NPs, T-Expand and Dynabeads™ was processed using Malvern DLS Zetasizer. Naked dextran NPs (10 μL, 0.75 mg/ml), T-Expand (10 μL, 0.75 mg/ml) and Dynabeads™ (2 μL) were dispersed in MiliQ water (2 ml) with gently sonicated water bath for 1 min. For each sample, three independent measurements were recorded and reported as mean ± SD ($n = 3$).

## Cryo-Transmission electron microscopy

Three μL of the sample was applied on a hydrophilized lacey carbon 300 mesh copper grid (Ted Pella Inc., California, USA). The excess sample on the grid was blotted with filter paper at blotting time of 3 s, blotting force 0, temperature 4 °C, and 100% humidity (FEI Vitrobot IV, Eindhoven, The Netherlands), and was rapidly plunged into liquid-nitrogen cooled ethane (−180 °C). Sample observations were performed using a Tecnai G2 20 transmission electron microscope (FEI, Eindhoven, The Netherlands) at a voltage of 200 kV under a low-dose rate. Images were recorded with an FEI Eagle camera 4k × 4k at a nominal magnification of 29k×.

## Quantifying the Azido molar on the surface of Dextran NPs

The sulfo-Cyanine5 DBCO (1 mg, cat 133F0, Lumiprobe) was dissolved in 1 mL of DMSO. The concentration calibration curve was obtained by detecting the servals concentration (0.25 μg/ml, 0.5 μg/ml, 0.75 μg/ml, 1 μg/ml, 2.5 μg/ml) of sulfo-Cyanine5 DBCO. Next, 10 μl of dextran NPs were incubated with 1 μl of sulfo-Cyanine5 DBCO (1 mg/ml) overnight. The NPs were pelleted at the bottom of the centrifuge tube by centrifugation at 10600 g for 10 minutes. The supernatant of un-clicked sulfo-Cyanine5 DBCO was measured by plate reader (Synergy H1, Biotek). The amount of clicked sulfo-Cyanine5 DBCO was determined by calculating the concentration difference before and after coupling.

## Fluorometric analysis of antibody conjugation

First, the anti-CD3 (clone OKT-3, BioXCell) and anti-CD28(clone 9.3, BioXCell) antibodies were dissolved in PBS buffer (0.5 mg/ml, 200 μl), Next, DBCO-Sulfo-NHS (3.32 μl, 10 mm) and Cy5-NHS (8 μl, 1.5 mM in DMSO, Cytiva, PA15101) was added into anti-CD3 solution, DBCO-Sulfo-NHS (3.32 μl, 10 mm) and Alexa Fluor (AF) 488-NHS (8 μl, 1.5 mM in DMSO, Thermo Fisher Scientific) was added into anti-CD28 solution and further incubated for 3 h at room temperature. After incubation, the DBCO-Cy5-anti-CD3 and DBCO-AF 488-anti-CD28 was purified by using Zeba spin columns (500 μl, 40 kDa MWCO) according to the manufacturer's protocol. In order to remove as much unreacted linker (DBCO-Sulfo-NHS, Cy5-NHS and AF 488-NHS) as possible, we purified the labeled antibody separately using two pre-equilibrated Zeba spin columns. For detailed procedures, see Figure Supplementary Fig. 10a. To construct precise calibration curves relating antibody fluorescence intensity to concentration, we combined 150 μL of DBCO–Cy5–anti-CD3 (0.0283 mg/mL) with 50 μL of DBCO–AF488–anti-CD28 (0.0283 mg/mL) to prevent spectral crosstalk between fluorophores, and from this mixture established individual calibration curves for anti-CD3 and anti-CD28.

To obtain a more accurate estimate of the antibody density on the nanoparticle surface, a single bifunctional tag strategy was employed. Anti-CD28 was first reacted with Cy5-DBCO-NHS (3.32 μl, 10 mM in DMSO, Broadpharm), thereby avoiding competition between fluorophore-NHS and DBCO-Sulfo-NHS reagents Supplementary Fig. 10c. A standard calibration curve for Cy5-DBCO–anti-CD28 was

generated by measuring fluorescence intensities of known concentrations in a microplate reader. Nanoparticle conjugation reactions were performed as described in Supplementary Fig. 10d, after which the supernatants were collected. Fluorescence measurements of these supernatants were converted to antibody mass using the calibration equation.

## Primary T-cell isolation and activation

Peripheral blood mononuclear cells (PBMCs) were sourced from healthy donors from Rigshospitalet, Copenhagen as approved by the Danish ethics committee. PBMCs were isolated from whole blood by density centrifugation using SepMate tubes (StemCell) and frozen at −80 °C in fetal bovine serum (FBS) (S1810, BIOWEST) and 10 % DMSO.

Primary T cells from healthy donor PBMCs were isolated using the EasySep™ Human T cell enrichment kit supplied by STEMCELL technologies (as per supplier instructions). The purified T cells were resuspended in $0.5 \times 10^6$ cells/ml in Roswell Park Memorial Institute (RPMI 1640) + 10% FBS + 1% penniciln streptomycin (Penstrep) (Thermo Fischer Scientific) and IL2 (20IU/μl) (PeproTech®). For T cells activation, 100,000 purified T cells were plated in 100 μL culture medium in a 96-well cell culture plate, then specific concentration (0.75 mg/ml: 1 μl, 2 μl, 5 μl, 10 μl) of T-Expand (or) Dynabeads™ at a bead-to-cell ratio of 3:1, (or) TransAct™ were added to each well and incubated in a humidified $CO_2$ incubator at 37 °C for 20 h. Next, the activated T cells were harvested and washed with PBS buffer for further flow cytometry analysis. The T cell activation is measured by staining the T cell activation markers CD69 and CD25. The expression of CD69 and CD25 was evaluated as the percentage of fluorochrome-labelled cells within the total cell population, using an LSRFortessa™ flow cytometer (BD Biosciences, USA).

For T cell expansion, purified T cells were seeded at a density of $1 \times 10^6$ cells/mL in a 48-well culture plate (or other suitable tissue culture plates/flasks) supplemented with rhIL-2 (20 IU/mL). Subsequently, 20 or 40 μL of T-Expand (0.75 mg/mL), Dynabeads™ at a bead-to-cell ratio of 3:1, or TransAct™ was added. Cultures were maintained in a humidified $CO_2$ incubator at 37 °C. The T cells were counted on day 3, 5, and 7, and their viability was reassessed using LUNA-FL™ Dual Fluorescence Cell Counter. The cells are resuspended in fresh media to maintain a concentration of $1 \times 10^6$ cells/mL, and rhIL-2 is replenished to a final concentration of 20 IU/mL to support continued growth and expansion. On Day 10, the culture contained Dynabeads™ was thoroughly resuspended to ensure uniform mixing of cells and beads which was then removed using a magnetic separator (DynaMag-15 magnetic rack, Thermo Fisher Scientific), while the T-Expand expanded CAR T cells, were washed with fresh cell medium and further transferred to a clean container. Finally, the cells are cryopreserved for downstream application.

## Cell lines and culture

The human cell lines Jeko-1(B cell lymphoma) and Jurkat obtained from American Type Culture Collection (ATCC) and cultured in RPMI 1640 supplemented with 10% FBS and 1% Penstrep. For cytotoxicity assay and in-vivo tumor models, these cell lines were transduced with mCherry fluorescent protein and luciferase genes using lentiviral vectors and were sorted on a FACS melody (BD Biosciences, USA) to establish fresh cultures.

## Biocompatibility assay of T-Expand

Jurkat T cells were co-incubated with a broad range of either naked dextran NPs or T-Expand at different doses (0.75 μg, 1.875 μg, 3.75 μg, 7.5 μg, 15 μg and 75 μg) and seeded into 96-well plates at $5 \times 10^5$ cells per well in 200 μl RPMI 1640 supplemented with 10% FBS and 1% Penstrep. Cell viability was assessed with the NucleoCounter NC-200 (Chemometec A/S, Denmark) on days 1, 4, 7, and 10, following the manufacturer's protocol.

## Airyscan microscope imaging of T-Expand interacting with Jurkat cells

The engineered Jurkat cells were cultured with T-Expand or Dynabeads™ for 24 hours. Following incubation, centrifugation was conducted to remove the culture medium and resuspend the cells in FACS buffer (PBS + 1% BSA). Cell membrane was stained with anti-human CD45 and incubated on ice for 20 min. The cells were subsequently collected by centrifugation and 5 μl of the cell suspension was transferred to a glass slide (Thermo Fisher Scientific). The cell droplet was mounted with a coverslip and the edges sealed with nail polish. Confocal images were obtained by using Airyscan confocal microscope (LSM 900 Airyscan 2, Carl Zeiss Microscopy GmbH). For real-time imaging, engineered Jurkat cells were cultured in an 8-well Glass μ-slide chamber (ibidi) at 37 °C and stimulated with T-Expand or Dynabeads™ for 24 h. 3D Z-stack imaging was conducted using an Airyscan laser confocal microscope, with subsequent analysis performed using Imaris software (v.10.1.0).

## CAR construct and Lentiviral production

A second-generation CAR were designed with an anti-CD19 (FMC63) CD8TM/hinge domain, 4-1bb, and CD3ζ. Expression of the CAR was linked with expression of a green fluorescent protein (GFP) reporter gene. The CAR was inserted into a third-generation lentiviral vector under the control of a human EF1α promoter. All plasmids were synthesized by GenScript, USA. Lentiviral particles were generated in HEK-293T cells by Lipofectamine transfection using packaging plasmids pRSV.REV (Addgene plasmid #12253), pMDLg/p.RRE (Addgene plasmid #12251), and pMD2.G (Addgene plasmid #12259), all gifted by Didier Trono[47], along with a transfer plasmid, a modified version of pLenti-puro (Addgene plasmid # 39481) gifted by Ie-Ming Shih[48], containing the CAR construct of interest. The transfected HEK-293T cells were incubated at 37 °C and 5% $CO_2$ and the cell culture supernatant were collected at 24 and 48 h after transfection. Collected supernatants were up concentrated for lentiviral particles using Lenti-X™ Concentrator according to the maufacturer's recommendations (Takara bio).

The titter of the produced lentivirus particles was assessed through a titration experiment by infecting SUP-T1 cells and subsequently analyzing GFP or surface expression of the receptor via tetramer staining and flow cytometry.

## Engineering of Jurkat with mCherry-Cathpesin L

A cathepsin L-mCherry fragment was synthesized and cloned into transfer vector by GenScript, USA. Lentivirus was created following the same protocol as the previous setup, and Jurkats were transduced using an MOI of 5, followed by sorting of the cells based on mCherry expression.

## Production of CAR T cells in vitro

On Day 0, the purified human T cells from healthy donors were diluted to $1 \times 10^6$ cells/mL in RPMI 1640 media containing recombinant human IL-2 (rhIL-2) at a final concentration of 20 IU/mL, which was then cultured with T-Expand (0.75 mg/ml, 20 ul), TransAct™ (10 μl) or Dynabeads™ (Gibco) based on a bead to cell ratio, typically 3:1 for 24 h. Then the CD19 CAR lentiviral particles were added to the culture at an MOI of 3 to ensure efficient transduction. Further the CAR T cells were expanded using the T cell expansion protocol.

## In vitro T-cell phenotype analysis

The phenotype of T cells and CAR T cells was evaluated using flow cytometry. The transduction efficiency of CAR was determined by the co-expression of two markers: the intracellular fluorescent protein GFP encoded by the CAR construct and CD19 tetramer. Flow cytometry staining was performed on ice for 20 min in FACS buffer. The concentration of antibodies used for staining was determined according

to the manufacturer's instructions. Stained cells were analyzed using an LSRFortessa flow cytometer, and data analysis was conducted using FlowJo v10 software. Cell lines were sorted on a FACSAria™ flow cytometer (BD Biosciences, USA) using a 100 μm nozzle. Amplitude was adjusted to optimize droplet break-off, and droplet calibration was performed before each sort using AccuDrop™ beads (BD, USA). Gates were set for each time point and sample independently based on fluorescence minus one (FMO) control. Cells were centrifuged at $500 \times g$ for 5 min at 4 °C, and the supernatant was discarded. The pelleted cells were resuspended in 5 μL of 1 μM Dasatinib (LC Laboratories, D3307), antigen tetramers were added, and the cells were incubated for 15 min at 37 °C in the dark. The cells were washed once with FACS buffer, followed by staining with Near-IR 28 (NiR) viability dye (Invitrogen™, L34976) and additional antibodies for surface staining for surface markers 30 min at 4 °C in the dark. Finally, the cells were washed twice in 200–300 μL of FACS buffer, and immediately analyzed on a LSRFortessa™ flow cytometer. Alternatively, the cells were fixed with 50 μL of 1% paraformaldehyde for 1–2 h (if required), washed twice with FACS buffer, and analyzed 2–24 h later.

Fluorochrome-conjugated antibodies including BB700-anti-human CD25 (IL-2R) (Clone M-A251), BV786-anti-human CD69 (Clone FN50), BV421-anti-human CD3 (Clone SP34-1), BV711-anti-human CD45RA (HI100), APC-anti-human CD197 (CCR7) (Clone 2-L1-A), BUV737-anti-human 279 (PD-1) (Clone EH12.1), BV650-anti-human CD223 (LAG-3) (Clone 11C3C65), BV786-anti-human CD39 (Clone TU66), BV605-anti-human TIGHT (Clone 741182), BV480-anti-human CD4 (Clone L200), PerCP-anti-human CD8 (Clone SK1), BV480-anti-human CD8 (Clone RPA-T8), BUV737-anti-human CD28 (Clone CD28.2), PE-anti-human CD137 (Clone 4B4-1), BV711-anti-CD19 (Clone SJ25C1), PE-Cy7-anti-CD4 (Clone RPA-T4), PE/Cy7-Streptavidin were purchased from BD Biosciences. PE-anti-human CD69 (Clone FN50), PE/Cy7-anti-human TIM-3 (Clone F38-2E2), PE-Cy7-anti-human CD57 (Clone QA17A04), BV605-anti-human CD27 (Clone O323) were purchased from Biolegend. The catalog numbers of all antibody and protein information was provided in Supplementary Table 3.

### Proliferation assay
The T cells were resuspended $1 \times 10^6$ cells/mL in PBS and 0,75ul/ml of CellTrace violet was added to the T cells and incubated at 37 °C for 15 min. After incubation and co-culturing with irradiated Jeko-1 target cells at a 1:5 effector to target ratio for 6 days, flow cytometry was used to measure CellTrace Violet dilution, and the proliferation index was calculated using FlowJo software.

### In vitro CAR T cell killing assays and rechallenge assay
The CD19 CAR T cells and the CD19 target cells, Jeko 1 lymphoma cell line modified to express mCherry, was resuspended in RPMI 1640 supplemented with 10%FBS and 1% Penstrep and seeded at the indictaed E:T ratio. On Day 0, $1 \times 10^4$ target cells and $1 \times 10^4$ CAR T cells were co-cultured in a 96-well plate with 200 μL of T-cell culture medium. After a 48-h incubation, the supernatant was collected for ELISA test, followed by the addition of another $1 \times 10^4$ target cells to each well. This process was repeated every 48 hours to simulate the repeated in vivo challenge faced by CAR T cells.

### In vivo efficacy studies
For in vivo studies, 6-week-old female NXG Immunodeficient mice (NOD-Prkdcscid-IL2rgTm1/Rj) were acquired from Janvier Labs and housed at the Bio Facility, Department of Health Technology, Technical University of Denmark. The mice were housed separately from the immunocompetent mice. The mice were housed in groups of 4–5 mice per cage, and the feeding was replenished every 3 days or as required. All procedures were approved by the Danish National Animal Experiment Inspectorate and the institutional ethical board (Approval no. 2020-15-0201-00748).

For efficacy studies, $1 \times 10^6$ Jeko-1 lymphoma cells expressing luciferase were injected intravenously into 7–10-week-old NXG mice. Tumors were allowed to engraft for 7 days before randomization and treatment start. On treatment day, indicated amount of untransduced, Dynabeads™-expanded T cells or T-Expand expanded CD19 CAR T cells in 200 μl PBS were administered i.v. by tail vein injection. Tumor growth was monitored weekly with bioluminescence imaging (Optical Imaging unit, MILabs). Mice received 30 mg/mL of D-Luciferin intraperitoneally and 20 min later bioluminescence was measured. Measurements were analyzed using fixed-size regions of interest using MILabs optical imaging software. Mouse body weight and appearance were monitored throughout the experiment.

At study termination, spleen and bone marrow were harvested for further analysis by flow cytometry. Spleens and bone marrow were passed through 70 μm filters and washed in PBS to create a single-cell suspension. Splenocytes subsequently went through red blood cell lysis before being passed through a 70 μm filter again. Blood was harvested on day 21 from the saphenous vein. Red blood cells were lysed before washing with PBS. Blood, splenocytes and bone marrow cells were surfaced stained for Near-IR 28 (NiR) viability dye and antibodies against surface markers, CD3, CD4, CD8, CD19, CD137, CD69 and CD19 CAR tetramer and run on LSRFortessa™ for flow cytometry analysis.

### Differential gene expression analysis and GSEA
Differential gene expression analysis was used to identify differentially expressed genes in PBMCs of four donors expanded with either T-Expand or Dynabeads™. Raw RNA-Seq reads were trimmed using Trim Galore and transcript abundances were quantified using Kallisto Quant[49]. The transcript abundances from Kallisto were imported into DESeq2[50] for differential expression analysis. Log2 Fold Change > 2 and <−2 with an adjusted $p$ value < 0.05 was used as threshold for over- and under-expressed genes for the analysis. The volcano plot was generated with ggplot2 and the heatmap was generated with Complex Heatmap[1].

Gene set enrichment analysis (GSEA) was performed using clusterProfiler[51,52] to identify significantly enriched Gene Ontology (GO) terms. As an input for the GSEA we used a ranked gene list of all significant genes ($p$.adj < 0.001) from the DESeq2 analysis, ordered by Log2 Fold Change. This ranked gene list was then used as input for the gseGO function from the clusterProfiler package to identify significantly enriched Gene Ontology (GO) terms. To retain only highly significant GO terms, this analysis was performed using minGSSize > 30 (selecting only gene sets with more than 30 genes) and $p$ value Cut off < 0.001.

### Statistical analysis
Data are represented as the mean ± standard deviation. When necessary, a two-tailed Student's $t$-test was used to identify statistically significant differences between the two groups, using GraphPad Prism 8 for computations. In instances of comparison among more than two groups, a one-way analysis of variance (ANOVA) was conducted, followed by Dunnet's multiple comparison test or Holm *Šídák's multiple comparison test*. Levels of statistical significance were denoted as follows: not significant, $P > 0.05$; *$P < 0.05$; **$P < 0.01$; ***$P < 0.001$.

### Reporting summary
Further information on research design is available in the Nature Portfolio Reporting Summary linked to this article.

## Data availability
The bulk RNA sequencing data have been deposited in the NCBI Gene Expression Omnibus (GEO) and accession numbers is GSE308384. The proteomics data have been deposited in the PRIDE repository via the ProteomeXchange Consortium under the identifier PXD067738. The

data supporting this study are provided in the main manuscript and the Supplementary Information. Additional data could be obtained from the corresponding author upon request. Source data are provided with this paper and are available for Figs. 1–6 and Supplementary Figs. 13, 14, 16, 17,18, 20, 21 in the associated Source Data file.

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

## Acknowledgements

This research was funded in part by: The Novo Nordisk Foundation, Challenge Programme 2021 —Smart Nanomaterials for Applications in Life-Science (grant no. 0066562) (to S.R.H., Y.S., and H.A.). A grant from the European Union's Horizon 2020 research and innovation program Marie Curie grant agreement No. 955575 (to S.R.H.)

## Author contributions

T.Z., K.R., M.O., Y.S., and S.R.H. designed the experiments. T.Z. contributed to T-Expand design and production. The in vitro and in vivo T cell assays were designed and carried out by K.R., T.Z., H.R.H., M.R.H., M.O., R.U.W.F. and C.R.P. The proteomics data was analyzed by H.L, and P.M.H.H. K.Q. and I.S. contributed to T-Expand characterizations. The bulk RNA sequencing data was analyzed by K.K.M. T.Z., K.R., and M.O. contributed to the organization and analysis of the data. T.Z., K.R., and M.O. co-wrote the paper. Y.S., S.R.H., and H.A. supported and supervised this project, co-wrote the paper. All authors discussed the results and revised the manuscript.

## Competing interests

The authors T.Z., K.R., M.O., H.R.H., Y.S., and S.R.H. are part of the patent application Dextran nanoparticles for T-cell activation and proliferation. European Patent Application No. 24178957.7. The other authors declare that they have no conflicts of interest.
