## [Transparent Peer Review file · Nature Communications]

Dextran-based T-cell expansion nanoparticles for manufacturing CAR T cells with augmented efficacy

Corresponding Author: Professor Sine Hadrup

Version 0:

Reviewer comments:

Reviewer #1

(Remarks to the Author)

The manuscript by Tao Zheng and colleagues presents a comprehensive and innovative study describing the development and evaluation of a novel dextran-based nanoparticle platform, termed T-Expand, for the ex vivo expansion of CAR T cells. According to the authors, this platform addresses key limitations of current artificial antigen-presenting cell (aAPC) systems such as Dynabeads™ and TransAct™, including overstimulation, poor scalability, and challenges in manufacturing. In the first part of the study, the authors design and characterize the T-Expand nanoparticles, which feature a high surface density of anti-CD3 and anti-CD28 antibodies conjugated via click chemistry to the modifiable side chains of dextran. They claim that the nanoscale size and biocompatibility of T-Expand eliminate the need for post-expansion removal, simplifying the manufacturing process and improving scalability.

In the second part of the study, the functional characteristics of CAR T cells expanded using T-Expand are assessed. The results suggest that these cells possess enhanced phenotypic and functional attributes, including a greater proportion of CD27⁺ and CD28⁺ effector memory T cells, reduced expression of exhaustion markers, and improved anti-tumor efficacy both in vitro and in vivo. Furthermore, bulk RNA sequencing reveals upregulation of effector genes, including cytokines, and chemokine receptors implicated in T cell trafficking and immune modulation. The authors propose that these transcriptional changes correlate with the superior in vivo persistence and anti-tumor performance observed in xenograft lymphoma models, where T-Expand-expanded CAR T cells outperformed higher-dose Dynabeads™-expanded cells.

This is a timely and overall well-executed study that introduces a promising new platform for CAR T-cell manufacturing. It represents a meaningful contribution to the field of adoptive cell therapy, with the potential to improve therapeutic efficacy and streamline production. However, before publication, the authors should address the following major and minor concerns:

Major concerns

1. The authors report the optimization of anti-CD3 and anti-CD28 antibody ratios used for the functionalization of the T-Expand particles. However, no data are provided regarding the final amount of each antibody successfully conjugated to the particles. Since conjugation efficiency may differ from the input ratios, quantification (e.g., using ELISA or fluorophore-tagged antibodies) is important to clearly report the final antibody composition present on the T-Expand particles used in the experiments.
2. While the authors emphasize the biocompatibility of the T-Expand nanoparticles as a key feature, no clear experimental data are provided to support this claim. The presented T cell proliferation data do not address the concentration-dependent effects of the particles on the overall cell population. In this context, I recommend including a biocompatibility assessment, such as cytotoxicity or cell proliferation assays using T-Expand particles at varying concentrations on T cells and/or other relevant cell lines, to confirm the system's safety profile.
3. Did the authors use equivalent concentrations of Dynabeads™, TransAct™, and T-Expand particles, considering their differing physical and biochemical nature? Specifically, do these platforms contain comparable amounts of anti-CD3 and anti-CD28 antibodies? Given that T-Expand is a nanoparticle-based system, while Dynabeads™ and TransAct™ are

microscale or polymer-based platforms, direct comparison may be confounded by differences in particle size, surface area, and antibody density. For example, was normalization performed based on the number of particles, the total antibody content, surface density of antibodies, or functional activity? The authors should clearly describe and justify the approach used to compare the different platforms and discuss whether the different concentrations used may lead to different results.

4. The authors claim that the particles do not require post-expansion removal due to their nanoscale size and biocompatibility. However, this assumes the particles remain stable and do not aggregate, accumulate, or degrade over time. I strongly recommend assessing the stability and integrity of T-Expand particles over relevant time points under physiological experimental conditions.

5. RNA-seq data indicate substantial upregulation of IFN- γ , pro-inflammatory cytokines (e.g., IL-6, IL-17), and other immune mediators in T-Expand-expanded CAR T cells. While these signatures may correlate with increased functionality, they also raise concerns about the potential for cytokine release syndrome (CRS) and immune effector cell-associated neurotoxicity syndrome (ICANS), two major clinical risks associated with CAR T-cell therapy. The authors should explicitly discuss these risks in the manuscript and consider whether the upregulated cytokine profiles observed for the T-Expand approach may pose translational safety concerns. In this context, it would be informative if the authors could report and discuss whether they observed any adverse reactions in their CAR T-cell experiments in mice.

6. Additional discussion is needed regarding the differing functional outcomes observed between the Dynabeads™ and T-Expand platforms. This is particularly important, as the two systems yield distinct clinical-relevant T cell activities and gene expression profiles. How do the authors explain the observed divergence in activity? Is the unique nature of the T-Expand nanoparticles, such as their size, flexibility, or nanoscale interaction dynamics, sufficient to account for these changes? Could the chemical composition of the dextran polymer itself contribute to altered T cell activation or downstream signaling? Alternatively, might these differences arise from distinct physicochemical properties of the particles, such as surface charge, stiffness, or antibody orientation and density? The manuscript currently lacks a detailed analysis and discussion regarding this point and a thoughtful discussion of these mechanisms would greatly strengthen the manuscript and support the claims made about the advantages of the T-Expand system over conventional aAPC platforms.

Other comments:

1. In lines 142–143, the authors state that T-Expand particles exhibit high uniformity in cryo-EM images. However, no quantitative analysis is presented from the cryoEM micrographs. Instead, the authors rely on NTA data showing a mean particle size of 199.8 ± 70.7 nm. Figure 1h shows a secondary peak at ~ 120 nm, suggesting the presence of a distinct subpopulation. The authors should discuss this heterogeneity and clarify whether the smaller peak represents unbound particles. If possible, quantify and report the proportion of each population.

2. Overall, the authors should provide more comprehensive descriptions of their experimental procedures, as the current level of detail is insufficient for reproducibility. For example, in lines 501–514, the section describing the formation of T-Expand particles for T cell activation and expansion lacks critical experimental information. This includes the concentrations of antibodies used, reaction volumes, buffer compositions, solvents, and technical details of the instrumentation employed. The authors are encouraged to thoroughly revise and complete all experimental sections. Specifically, the sources and references for the anti-CD3 and anti-CD28 antibodies should be clearly stated. Additionally, the origin of all cell lines used, along with their corresponding unique identifiers or commercial catalog numbers, should be provided in the methods to ensure traceability.

Reviewer #2

(Remarks to the Author)

This manuscript describes dextran nanoparticles to which T cell stimulating antibodies have been conjugated using click chemistry, and their use to promote T cell and CAR-T cell expansion, along with analysis of the function of the resultant cells. Overall, the authors provide a set of very nice studies characterizing the nanoparticles, and T cell expansion with these dextran-based nanoparticles. One major concern is the novelty of this work. A number of nanoparticle and microparticle artificial antigen presenting cells (aAPCs) have been developed and published, so the novelty of the concept is not high. The authors highlight the improvements, as versus Dynabeads, that they find when using these particles to expand T cells, particularly in relation to the functionality of the resultant cell populations. However, it is not clear the authors provide a fair comparison, as they only compare to a single Dynabead condition. The authors first optimize the dose of their nanoparticles to obtain optimal expansion and T cell phenotype, and then utilize that optimal dose for subsequent studies. The finding that stimulation dose is critical to expansion and function of the T cells is consistent with past literature using other aAPCs. So a different, optimized, dose of Dynabeads in these present studies may result in very similar, beneficial impact on CAR-T expansion, differentiation status, and function. Without a comparison of the optimized Dynabeads and optimized dextran nanoparticles, one cannot know if one provides a true advantage.

Technical question – the size distribution of the dextran nanoparticles have a much larger size after antibody conjugation than prior. How do the authors explain? Also, the size distribution post-antibody conjugation has two peaks; one approx. the same size as reported for the naked NPs, the other much larger – how do the authors account for this

Reviewer #3

(Remarks to the Author)

Remarks to the authors:

The manuscript by Zheng et al. presents a well-conceived and technically rigorous study that introduces "T-Expand," a novel dextran-based nanoparticle platform for T cell activation and CAR T cell manufacturing. The authors demonstrate that T-Expand enables robust T cell expansion, enhanced CD19 CAR T cell cytotoxicity, and in vivo tumor clearance at reduced cell doses. The biocompatibility, high antibody loading capacity, and simplified downstream processing of T-Expand make it a promising alternative to traditional expansion systems such as Dynabeads™. The manuscript is well-structured, the experiments are thorough, and the conclusions are generally well-supported. However, there are several areas where additional data, clarifications, or contextualization would significantly strengthen the claims and translational relevance.

Major issues:

1. The use of azido-dextran and DBCO click chemistry is well-established in immunomaterials literature (e.g., Broaders et al., PNAS, 2009; Adv Mater. 2017;29(7):1603239.). The manuscript should more clearly articulate what is unique about this specific implementation and cite foundational references to distinguish engineering novelty from repurposed techniques.
2. In the transmission electron microscopy (TEM) image shown in Figure 1g, the T-Expand nanoparticle on the left appears to exhibit a double-layered structure or possibly contains a smaller particle within it. This morphology is not addressed in the text and may lead to confusion regarding particle uniformity. The authors could either clarify this observation in the figure legend or main text, or consider replacing the image with a more representative example that better reflects the described nanoparticle population.
3. In Figure 5i, the bioluminescence images in row 2, columns 3 and 6 appear to be identical, despite representing different treatment groups. Please verify whether this is a case of inadvertent image duplication.

Minor issues:

1. The nanoparticle tracking analysis (NTA) data in Figure 1h suggests a potential bimodal distribution of particle size, with what appears to be two peaks. This raises the question of whether the T-Expand formulation consists of two distinct nanoparticle populations. The authors should elaborate on this in the Results section-either to explain the source of heterogeneity, or to clarify whether this is an artifact of measurement or a true feature of the formulation.
2. The representative contour plot in Supplementary Figure 16a appears to display an unusually low number of events, which limits the interpretability of the flow cytometry data. The authors should clarify the number of events collected and justify whether this sample is truly representative. If possible, consider replacing it with a plot from a higher-quality acquisition to support the conclusions drawn.

Version 1:

Reviewer comments:

Reviewer #1

(Remarks to the Author)

After a thorough evaluation of the revised version of the manuscript "Dextran-based Nanoparticle Platform (T-Expand) for CAR T-Cell Expansion" by Zheng et al. (Ref. NCOMMS-25-30476A), I am fully convinced that the authors have satisfactorily addressed all my previous major concerns.

I particularly appreciate the authors' efforts in performing new, well-designed experiments that directly respond to the reviewers' major critiques. The inclusion of quantitative stoichiometric analysis, detailed biocompatibility and stability assessments, and expanded methodological descriptions has considerably improved the quality of the manuscript. Furthermore, the revised discussion now provides a thoughtful and balanced interpretation of the translational relevance and safety considerations, including potential implications for cytokine release and functional divergence relative to existing aAPC platforms.

Overall, the manuscript now presents a comprehensive, methodologically sound, and compelling body of work. The additional data and clarifications convincingly demonstrate the novelty and translational value of the reported T-Expand platform. I am satisfied that all major concerns have been fully resolved, and I believe the manuscript is now suitable for publication in its current form.

Reviewer #2

(Remarks to the Author)

The authors have provided a variety of new/altered data and analysis in response to the original reviews. Unfortunately, however, they did not respond adequately to the main technical concern from the original review - a comparison of the optimized Dynabeads and optimized dextran nanoparticles is required to establish if this new platform provides a true advantage. The experiment here is quite straightforward - repeat previous experiments with a range of Dynabead:cell ratios, similarly to how the studies were performed with their particles, but the authors did not perform.

Reviewer #3

(Remarks to the Author)

The authors have fully addressed my prior concerns with appropriate additional data, figure corrections, and mechanistic clarification. Therefore I support the publication of this work.

Version 2:

Reviewer comments:

Reviewer #2

(Remarks to the Author)

The authors have appropriately addressed the earlier concern with the inclusion of new data addressing the question.

Point-by-Point Response to Reviewer's Comments (Zheng et al. formerly manuscript NCOMMS-25-30476)

For ease of reading, the reviewer's comments are in regular typeface, our responses are highlighted in italics

Point by point reply to Reviewer #1 (Remarks to the Author):

Reviewer #1 (Remarks to the Author):

The manuscript by Tao Zheng and colleagues presents a comprehensive and innovative study describing the development and evaluation of a novel dextran-based nanoparticle platform, termed T-Expand, for the ex vivo expansion of CAR T cells. According to the authors, this platform addresses key limitations of current artificial antigen-presenting cell (aAPC) systems such as Dynabeads™ and TransAct™, including overstimulation, poor scalability, and challenges in manufacturing.

In the first part of the study, the authors design and characterize the T-Expand nanoparticles, which feature a high surface density of anti-CD3 and anti-CD28 antibodies conjugated via click chemistry to the modifiable side chains of dextran. They claim that the nanoscale size and biocompatibility of T-Expand eliminate the need for post-expansion removal, simplifying the manufacturing process and improving scalability.

In the second part of the study, the functional characteristics of CAR T cells expanded using T-Expand are assessed. The results suggest that these cells possess enhanced phenotypic and functional attributes, including a greater proportion of CD27⁺ and CD28⁺ effector memory T cells, reduced expression of exhaustion markers, and improved anti-tumor efficacy both in vitro and in vivo. Furthermore, bulk RNA sequencing reveals upregulation of effector genes, including cytokines, and chemokine receptors implicated in T cell trafficking and immune modulation. The authors propose that these transcriptional changes correlate with the superior in vivo persistence and anti-tumor performance observed in xenograft lymphoma models, where T-Expand-expanded CAR T cells outperformed higher-dose Dynabeads™-expanded cells.

This is a timely and overall well-executed study that introduces a promising new platform for CAR T-cell manufacturing. It represents a meaningful contribution to the field of adoptive cell therapy, with the potential to improve therapeutic efficacy and streamline production. However, before publication, the authors should address the following major and minor concerns:

Major concerns

1. The authors report the optimization of anti-CD3 and anti-CD28 antibody ratios used for the functionalization of the T-Expand particles. However, no data are provided regarding the final amount of each antibody successfully conjugated to the particles. Since conjugation efficiency may differ from the input ratios, quantification (e.g., using ELISA or fluorophore-tagged antibodies) is important to clearly report the final antibody composition present on the T-Expand particles used in the experiments.

*Response: In the revised manuscript, we have incorporated a two-step fluorometric analysis to investigate the amount of conjugated antibody density on the surface of NP, now detailed in the **Results** sections and **Methods** (Figure 1f in revised manuscript, Supplementary Fig. 10 in revised supplementary materials), with all raw data provided in the **source data**. Details of the added data, is described below:*

Step A – Stoichiometric verification.

To confirm that the surface ratio of anti-CD3 to anti-CD28 matches the input ratio, anti-CD3 was labeled with Cy5-NHS and anti-CD28 with Alexa Fluor 488-NHS; both antibodies were simultaneously functionalized with DBCO-Sulfo-NHS (Figure S1, a). A microplate reader was used to quantify the Anti-CD3 and Anti-CD28 fluorescence calibration curves (Figure S1, b). Then, DBCO-Cy5-anti-CD3 and DBCO-Alexa Fluor 488-anti-CD28 at different concentration ratios (3:1; 1:1; 1:3) were subjected to click coupling reaction with naked NP. After the click-coupling reaction, NPs were pelleted by centrifugation. Fluorescence intensities of the supernatants were quantified against calibration curves for each antibody, allowing us to back-calculate the amount of antibody bound. The bounded anti-CD3:anti-CD28 ratio on the particle surface matched the input ratio within experimental error (Supplementary Fig. 10a, b in revised supplementary materials and added below).

Step B – Absolute loading determination.

To obtain a more accurate estimate of the amount of antibody on the nanoparticle surface, we first confirmed that the CD3:CD28 ratio on the particles matched the input ratio, because concurrent labeling of an antibody with both a fluorophore and DBCO-Sulfo-NHS can introduce competition between the two reagents and potentially underrepresent the total amount conjugated. We next adopted a single bifunctional tag. Specifically, anti-CD28 was derivatized with Cy5-DBCO-NHS, eliminating any competition and enabling direct fluorescence quantification. A calibration curve for Cy5-DBCO-anti-CD28 was generated with a microplate reader, and the resulting equation was applied to the supernatant measurements to calculate the mass of antibody bound per milligram of nanoparticles (Supplementary Fig. 10c, d in revised supplementary materials and added below). The final loading was $18.2 \pm 3.1 \mu\text{g}$ of

antibody per milligram of dextran nanoparticles (*Figure 1f in revised manuscript and added below*).

Together, these experiments demonstrate (i) that the final anti-CD3/anti-CD28 stoichiometry on T-Expand is controlled by the feed ratio, and (ii) that the platform attains an average antibody amount of $\sim 18 \mu\text{g mg}^{-1}$ NP. We believe these additions substantially strengthen the rigor and reproducibility of the study, and we are grateful for the reviewer's constructive recommendation. *The related description was provided in the result section, line 137 to 147 in revised manuscript.*

Supplementary Fig. 10. Quantification of conjugated antibody density on the surface of NP. The antibody conjugation efficiency on the surface of naked NPs measured by Nanodrop. **a**, Quantification of the conjugated anti-CD3 and anti-CD28 antibodies ratio on the surface of dextran NPs using a fluorescence spectroscopy microplate reader. **b**, anti-CD3 calibration curve used for final quantification and anti-CD28 calibration curve used for final quantification. **c**, Quantification of the conjugated anti-CD28 antibodies on the surface of dextran NPs using a fluorescence spectroscopy microplate reader. **d**, anti-CD28 calibration curve used for final quantification.

Figure 1f. The antibody conjugation efficiency on the surface of naked NPs measured by microplate reader.

2. While the authors emphasize the biocompatibility of the T-Expand nanoparticles as a key feature, no clear experimental data are provided to support this claim. The presented T cell proliferation data do not address the concentration-dependent effects of the particles on the overall cell population. In this context, I recommend including a biocompatibility assessment, such as cytotoxicity or cell proliferation assays using T-Expand particles at varying concentrations on T cells and/or other relevant cell lines, to confirm the system's safety profile.

Response: To verify the biocompatibility of our platform, we have added a dedicated biocompatibility study to the revised manuscript. Jurkat T cells were co-incubated with a broad range of either naked dextran nanoparticles or T-Expand and seeded into 96-well plates at 5×10^5 cells per well. Cell viability was assessed with the NucleoCounter NC-200 on days 1, 4, 7, and 10, following the manufacturer's protocol. Across all concentrations and time points, cell survival consistently exceeded 90%, demonstrating excellent biocompatibility for both naked NPs and T-Expand. The complete experimental protocol has been added as a new "Biocompatibility Assay of T-Expand" subsection in the **Methods** (line 696 to 701 in revised manuscript.) and related discussion has been inserted in the line of 162 to 164 in the revised manuscript. the quantitative data and statistical analyses are presented in the Results section (Supplementary Fig. 14 in revised supplementary materials and added below).

Supplementary Fig. 14. Quantification of cell viability (determined by NucleoCounter NC-200 machine) at day 1, 4, 7 and 10 by co-culturing naked NPs and T-Expand with Jurkat T cells.

3. Did the authors use equivalent concentrations of Dynabeads™, TransAct™, and T-Expand particles, considering their differing physical and biochemical nature? Specifically, do these platforms contain comparable amounts of anti-CD3 and anti-CD28 antibodies? Given that T-Expand is a nanoparticle-based system, while Dynabeads™ and TransAct™ are microscale or polymer-based platforms, direct comparison may be confounded by differences in particle size, surface area, and antibody density. For example, was normalization performed based on the number of particles, the total antibody content, surface density of antibodies, or functional activity? The authors should clearly describe and justify the approach used to compare the different platforms and discuss whether the different concentrations used may lead to different results.

Response: We thank the reviewer for highlighting the critical issue of dose normalization. Because the total antibody load, CD3:CD28 stoichiometry, and conjugation chemistry for Dynabeads™ and TransAct™ are proprietary, only limited information is publicly available, that Dynabeads™ display around 2.8×10^5 antibody molecules per bead with an anti-CD3:anti-CD28 molar ratio of ~3:1 [1], whereas

TransAct™ is reported to present a 1:1 ratio but without a disclosed antibody amount [2]. These commercial microscale carriers therefore differ markedly from our nanoscale T-Expand particles in diameter, specific surface area, and ligand density, making it challenging to normalize on the basis of absolute antibody site number.

We therefore adopted a clinically oriented comparison, The T-Expand and Dynabeads™ were seeded with the same initial number of T cells; the dose of Dynabeads™ (beads to cell ratio = 3:1) were used exactly according to commercial (thermo fisher scientific) recommendation for clinical CAR T cell manufacture protocol [3] and adopted by many research group [4-8] while T-Expand was tested across a 0.75µg to 30 µg dose of range T-Expand to identify the optimal concentration. The unified read-out was the fold expansion achieved after 10 days. The dose–response data show that T-Expand at ≥15 µg attains a cell-expansion efficiency comparable to Dynabeads™ used at a 3:1 bead-to-cell ratio (Figure 3c in revised manuscript, Figure S16 in revised supplementary materials). This design ensures the comparator was tested at its recommended clinical baseline while allowing a fair search for T-Expand's optimum.

4. The authors claim that the particles do not require post-expansion removal due to their nanoscale size and biocompatibility. However, this assumes the particles remain stable and do not aggregate, accumulate, or degrade over time. I strongly recommend assessing the stability and integrity of T-Expand particles over relevant time points under physiological experimental conditions.

Response: *Thank you for your attention to our statement that “no removal is required.” By this, we mean that T-Expand, due to their nanoscale dimensions, are effectively eliminated during standard T-cell washing procedures, obviating the magnetic-separation step necessary for Dynabeads™. In support of this, We conducted an additional experiment in which T-Expand was incubated with T cells for varying durations, followed by centrifugal washing. TEM imaging of the washed T cells was then performed to assess whether T-Expand remained on the cell surface or in the suspension, thereby demonstrating that T-Expand can be effectively removed by standard washing procedures. The TEM imaging confirmed the absence of T-Expand on the T-cell surface or in the post-wash suspension (Figure S15 in revised supplementary materials and inserted below). To further substantiate this attribute and assess nanoparticle stability under physiologically relevant conditions, we have incorporated the following two experimental series.*

First, supernatants were collected at various time points following T-cell expansion with either T-Expand (15 µg/mL) or Dynabeads™ (75 µL/mL, 3:1 beads to cell ratio). These supernatants were then diluted 1:1 and applied to fresh, unactivated primary T cells, and activation was measured 20 hours later by CD69/CD25 co-expression. The stimulatory potency of the T-Expand supernatant declined progressively after day 6 (Figure 1i in revised manuscript and inserted below). To eliminate any cell-mediated contributions, we incubated T-Expand and Dynabeads™ in cell-free RPMI 1640 (10%

FBS, 37 °C, 5% CO₂), collected the media on days 2, 6, and 10, and used them directly to stimulate primary T cells. Under these acellular conditions, T-Expand lost significant activity by day 6, whereas Dynabeads™ retained a relatively stable stimulatory effect (Figure 1j in revised manuscript and inserted below). These data indicate that the active moieties of T-Expand are gradually lost over time. This means, that in a clinical use scenario, even residual, inadequately washed T-Expand fails to activate T cells. The experimental design, results, and corresponding discussion have been added to the revised manuscript to address the concerns related to stability, and need for removal of T-Expand. The related discussion has been added in the line of 165 to 179 in the revised manuscript.

Supplementary Fig. 15. Representative transmission electron microscopy (TEM) images of primary T cell and activated at different time point (15 mins, 30 mins, 120 mins and 24 h). Scale bars: 5 μ m.

i**j**
Figure 1. *i*, the T cell activation profile in terms of % of T cells expressing CD69+CD25+ after 20 hrs of co-incubation with the supernatant obtained from the expansion cultures at different time point. *j*, The T-Expand (20 μ l) and Dynabeads™ (75 μ l) were incubated in Roswell Park Memorial Institute (RPMI) 10%FCS at 37°C and 5% CO₂ culture conditions for 10 days and the media was removed at day 2,6 and 10 timepoints of culture and was added to unactivated T cells at 1:1 dilution. Data in panels *i*, *j* represented as mean \pm SD, of biological replicates with each dot indicating one healthy donor sample(n=1).

5. RNA-seq data indicate substantial upregulation of IFN- γ , pro-inflammatory cytokines (e.g., IL-6, IL-17), and other immune mediators in T-Expand-expanded CAR T cells. While these signatures may correlate with increased functionality, they also raise concerns about the potential for cytokine release syndrome (CRS) and immune effector cell-associated neurotoxicity syndrome (ICANS), two major clinical risks associated with CAR T-cell therapy. The authors should explicitly discuss these risks in the manuscript and consider whether the upregulated cytokine profiles observed for the T-Expand approach may pose translational safety concerns. In this context, it would be informative if the authors could report and discuss whether they observed any adverse reactions in their CAR T-cell experiments in mice.

Response: We thank the reviewer for this important and clinically relevant comment. We agree that the upregulation of IFN- γ and pro-inflammatory cytokines, such as IL-6 and IL-17, observed in our RNA-seq analysis could, in principle, be associated with cytokine release syndrome (CRS) or immune effector cell-associated neurotoxicity syndrome (ICANS) in the clinical setting. In the revised Discussion section, we now explicitly address these potential translational safety concerns. Related discussions are provided *in line from 540 to 553 in revised manuscript and added below:*

“There is marked upregulation of pro-inflammatory cytokine transcripts (IFN- γ , IL-6, IL-17) in T-Expand–expanded CAR T cells relative to Dynabeads™. While such signatures are consistent with heightened effector programming, they overlap with pathways implicated in clinical cytokine release syndrome (CRS) and immune-effector cell–associated neurotoxicity syndrome (ICANS) [9]. Importantly, bulk transcriptomic elevation does not equate to systemic cytokine release, and in patients the surge in IL-6 during CRS is predominantly mediated by monocyte/macrophage activation rather than by T cells themselves [10, 11].

In our NSG mouse xenograft experiments, we did not observe any adverse reactions, including distress, weight loss, behavioral changes, during the treatment period, despite the potent antitumor activity of T-Expand–expanded CAR T cells. Considering that NSG mice are not an appropriate system to model CRS and ICANS, this should be evaluated in future studies using immune-competent and humanized animal models in which myeloid-driven cytokine biology is preserved”.

6. Additional discussion is needed regarding the differing functional outcomes observed between the Dynabeads™ and T-Expand platforms. This is particularly important, as the two systems yield distinct clinical-relevant T cell activities and gene expression profiles. How do the authors explain the observed divergence in activity? Is the unique nature of the T-Expand nanoparticles, such as their size, flexibility, or nanoscale interaction dynamics, sufficient to account for these changes? Could the chemical composition of the dextran polymer itself contribute to altered T cell activation or downstream signaling? Alternatively, might these differences arise from distinct physicochemical properties of the particles, such as surface charge, stiffness, or antibody orientation and density? The manuscript currently lacks a detailed analysis and discussion regarding this point and a thoughtful discussion of these mechanisms would greatly strengthen the manuscript and support the claims made about the advantages of the T-Expand system over conventional aAPC platforms.

Response: *Based on the activation assay, we confirm that neither naked dextran nanoparticles nor the azide-modified polymer backbone activate T cells or alter downstream signaling (Figure 1e in the revised manuscript and inserted below). We also show that T-Expand and Dynabeads™ have similar surface zeta potentials (Figure 1d in the revised manuscript and inserted below), and although Atomic Force Microscopy (AFM) imaging measurements reveal that T-Expand’s surface Young’s modulus (488 Kpa, Figure 1 and table 1 shown below) is much lower than Dynabeads™ (20–40 MPa), both platforms still effectively expand T cells, implying stiffness is not a decisive factor here. Although we lack orientation data for Dynabeads™ or TransAct™, the fact that T-Expand retains high stimulatory potency after serum incubation (Figure S12,13 in revised supplementary materials) indicates that antibody orientation is not limiting on our platform. Related discussions are provided in line from 489 to 506 in revised manuscript and added below:*

“Mechanistically, the divergent functional outcomes of T-Expand and Dynabeads™ appear to be driven by differences in stimulus intensity and kinetics. T-Expand presents a higher surface density of anti-CD3/anti-CD28, delivering more activation signals and enabling the rapid formation of multiple immunological synapses. This brief, high-density “burst” is sufficient to initiate T-cell activation and proliferation without prolonged high-strength engagement. By contrast, we observed that Dynabeads™ remain tightly connected to T-cell membranes throughout the expansion period, providing continuous strong signaling that while promoting early proliferation, may also predispose cells to activation-induced exhaustion. In addition, the size and biocompatibility of T-expand facilitates receptor-mediated internalization, which aligns with the natural T cell behavior upon TCR activation. We hypothesize that the internalization capacity contributes to the observed phenotype.

Consistent with this model, our stimulation-stability assays show that, under cell-free conditions, T-Expand’s stimulatory activity declines by day 6, indicating natural dissipation/degradation that limits prolonged overstimulation, whereas Dynabeads™ maintain relatively constant potency. In line with this, CAR-T cells expanded with Dynabeads™ exhibit higher exhaustion-marker expression in a three-round in vitro tumor rechallenge assay, whereas those expanded with T-Expand retain a less-exhausted CD27⁺CD28⁺ phenotype and superior durability under repeated challenge”.

Figure 1. d, Zeta potential of naked NPs, T-Expand and Dynabeads™ (n=3). **e,** Percent of T cells expressing the activation markers CD25 and CD69 after co-incubation with T-Expand (number of donors: n = 3).

Figure 1. AFM topography image of the particles.

Table 1. Nanomechanical properties measured by AFM

Sample	Young Modulus (KPa)
T-Expand	488
Dynabeads™	20000-40000 [1]

Other comments:

1. In lines 142–143, the authors state that T-Expand particles exhibit high uniformity in cryo-EM images. However, no quantitative analysis is presented from the cryoEM micrographs. Instead, the authors rely on NTA data showing a mean particle size of 199.8 ± 70.7 nm. Figure 1h shows a secondary peak at ~ 120 nm, suggesting the presence of a distinct subpopulation. The authors should discuss this heterogeneity and clarify whether the smaller peak represents unbound particles. If possible, quantify and report the proportion of each population.

Response: In accordance with the suggestion, we re-imaged T-Expand by cryo-TEM. The resulting mean diameters for T-Expand were 173.1 ± 37.27 nm as quantify by image J. These data, now presented as histograms with population percentages in figure 1h in revised manuscript and inserted below.

Figure 1. h, The size distribution of T-Expand, as determined by Cryo-TEM and measured by software of image J with 3 independent figures.

Additionally, we do not attribute peak of 120 nm to unconjugated particles. Normally, the strain-promoted alkyne–azide cycloaddition (SPAAC) between azide-decorated NPs and DBCO-antibodies proceeds with high efficiency, and NanoDrop spectroscopy at 309 nm confirms successful incorporation of DBCO into the antibody preparation (Figure S9 in revised supplementary materials). Owing to the high efficiency of the strain-promoted alkyne–azide cycloaddition between azide-functionalized nanoparticles and DBCO-modified antibodies, the ~120 nm peak does not represent nanoparticle populations lacking antibody conjugation.

Finally, we repeated the NTA measurements and investigated the previously observed subpopulation peak. We identified minor contaminants in the original NTA microfluidic chip. After thorough cleaning of the chip and re-analysis, the subpopulation peak was no longer detected (Figure S11a in revised supplementary materials and added below). Concurrently, we performed DLS-based particle sizing as a complementary control (Figure S11b in revised supplementary materials and inserted below), which yielded an average T-Expand particle diameter of approximately 200 nm and consistent with the NTA results. The size measured by NTA and DLS is large than Cryo-TEM imaging, this is because the physical diameter of the NP (as determined by TEM imaging) is typically smaller than its hydrodynamic diameter.

a

b

Supplementary Fig. 11. a, The size distribution of T-Expand, measured by NTA. **b,** The size distribution of T-Expand, measured by DLS.

2. Overall, the authors should provide more comprehensive descriptions of their experimental procedures, as the current level of detail is insufficient for reproducibility. For example, in lines 501–514, the section describing the formation of T-Expand particles for T cell activation and expansion lacks critical experimental information. This includes the concentrations of antibodies used, reaction volumes, buffer compositions, solvents, and technical details of the instrumentation employed. The authors are encouraged to thoroughly revise and complete all experimental sections. Specifically, the sources and references for the anti-CD3 and anti-CD28 antibodies should be clearly stated. Additionally, the origin of all cell lines used, along with their corresponding unique identifiers or commercial catalog numbers, should be provided in the methods to ensure traceability.

Response: We thank the reviewer for raising this point. We have revised the **Methods** section throughout to ensure that details are provided. Specifically, we have expanded the description of T-Expand particle preparation and T-cell activation to include all

concentrations, reaction volumes, buffer compositions, solvents, and relevant instrument specifications. We have also provided the sources, catalog numbers, and unique identifiers for anti-CD3 and anti-CD28 antibodies, as well as the origin and catalog information for all cell lines used. These additions ensure full traceability and reproducibility of our experiments.

Point by point reply to Reviewer #2 (Remarks to the Author):

This manuscript describes dextran nanoparticles to which T cell stimulating antibodies have been conjugated using click chemistry, and their use to promote T cell and CAR-T cell expansion, along with analysis of the function of the resultant cells. Overall, the authors provide a set of very nice studies characterizing the nanoparticles, and T cell expansion with these dextran-based nanoparticles. One major concern is the novelty of this work. A number of nanoparticle and microparticle artificial antigen presenting cells (aAPCs) have been developed and published, so the novelty of the concept is not high. The authors highlight the improvements, as versus Dynabeads, that they find when using these particles to expand T cells, particularly in relation to the functionality of the resultant cell populations. However, it is not clear the authors provide a fair comparison, as they only compare to a single Dynabead condition. The authors first optimize the dose of their nanoparticles to obtain optimal expansion and T cell phenotype, and then utilize that optimal dose for subsequent studies. The finding that stimulation dose is critical to expansion and function of the T cells is consistent with past literature using other aAPCs. So a different, optimized, dose of Dynabeads in these present studies may result in very similar, beneficial impact on CAR-T expansion, differentiation status, and function. Without a comparison of the optimized Dynabeads and optimized dextran nanoparticles, one cannot know if one provides a true advantage.

Technical question – the size distribution of the dextran nanoparticles have a much larger size after antibody conjugation than prior. How do the authors explain? Also, the size distribution post-antibody conjugation has two peaks; one approx. the same size as reported for the naked NPs, the other much larger – how do the authors account for this

Response: *We thank the reviewer for their positive comments related to our work. We have addressed each comment in a point-by-point manner.*

*First of all, based on the novelty concerns, while various nanoparticle- and microparticle-based aAPC platforms have been reported, T-Expand offers three key innovations that substantially enhance its novelty and translational potential (shown below). **Related discussions are provided in line from 455 to 465 in revised manuscript.***

1. High-Density Ligand Loading.

It is well established that optimal T-cell activation and proliferation depend on parameters such as particle size, shape, ligand density, and rigidity. Micron-scale particles typically excel at T-cell stimulation because their larger contact area promotes more immunological synapse formation and greater ligand–receptor pairing. However, to achieve comparable activation efficiency at the nanoscale, aAPCs must present very high ligand densities within their limited contact footprint. Existing polymer-based systems, such as PLGA and PEG-PDLLA nanoparticles, only introduce functional groups at chain termini, resulting in sparse surface click-handles after self-assembly.

In contrast, our azide-functionalized dextran features dense azido groups along every side chain, dramatically increasing antibody loading per particle. This high ligand density enables powerful T-cell activation and expansion at the nanoscale. Moreover, we have carried out extensive functional studies of CD19 CAR-T cells expanded with T-Expand, demonstrating enhanced cytotoxicity, persistence, and phenotype both in vitro and in vivo.

2. One-Step Bioorthogonal Conjugation.

Most reported aAPC platforms ranging from inorganic beads to polymeric microparticles which require multiple surface modification steps and secondary linkers to attach stimulatory antibodies [1, 12, 13]. By contrast, azide-modified dextran nanoparticles react directly with DBCO-anti-CD3 and DBCO-anti-CD28 in a single strain-promoted alkyne-azide cycloaddition (“click”) step. This one-pot process eliminates the need for additional activation or coupling chemistry, greatly simplifying manufacturing and reducing batch variability.

Figure 1. Reference of related aAPCs. **a.** alginate microgel based aAPCs. **b.** graphite micro-sheets based aAPCs. **c.** Silica microrod based aAPCs.

3. Beyond T-Cell Stimulation: Impact of T-Expand on CAR T-Cell Expansion and Function

In addition to the general advances of nanoscale aAPCs, we would like to emphasize that our study goes beyond T-cell stimulation alone. Specifically, we extend the application of the T-Expand platform into the clinically critical context of CAR T-cell manufacturing. We provide an in-depth analysis of how T-Expand-expanded CAR T cells differ in phenotype, persistence, and anti-tumor efficacy. To our knowledge, very few reports of nanoscale aAPCs have systematically investigated their impact on CAR T-cell expansion and downstream therapeutic performance. This extension into CAR T-cell production and functional evaluation further distinguishes our work and underscores its translational relevance.

Without a comparison of the optimized Dynabeads and optimized dextran nanoparticles, one cannot know if one provides a true advantage.

Response: *We thank the reviewer for highlighting the optimized dose comparison. Because the total antibody load, CD3:CD28 stoichiometry, and conjugation chemistry for Dynabeads™ and TransAct™ are proprietary, only limited information is publicly available. According to the reported work that Dynabeads™ display $\approx 2.8 \times 10^5$ antibody molecules per bead with an anti-CD3:anti-CD28 molar ratio of $\sim 3:1$ [1], whereas TransAct™ is reported to present a 1:1 ratio but without a disclosed antibody amount [2]. These commercial microscale carriers therefore differ markedly from our nanoscale T-Expand particles in diameter, specific surface area, and ligand density, making it challenging to normalize on the basis of absolute antibody site number.*

We therefore adopted a clinically oriented comparison, The T-Expand and Dynabeads™ were seeded with the same initial number of T cells; the dose of Dynabeads™ (beads to cell ratio = 3:1) were used exactly according to commercial (thermo fisher scientific) recommendation for clinical CAR T cell manufacture protocol [3] and adopted by many research works [4-8] while T-Expand was tested across a 0.75 μg –30 μg dose of range T-Expand to identify the optimal concentration. The unified read-out was the fold expansion achieved after 10 days. The data of dose–response data show that T-Expand at $\geq 15 \mu\text{g}$ attains a cell-expansion efficiency comparable to Dynabeads™ used at a 3:1 bead-to-cell ratio (Figure 3c in revised manuscript, Figure S16 in revised supplementary materials). This design ensures the comparator was tested at its recommended clinical baseline while allowing a fair search for T-Expand's optimum.

Technical question – the size distribution of the dextran nanoparticles have a much larger size after antibody conjugation than prior. How do the authors explain?

Response: We attribute the apparent increase in particle size after antibody conjugation to two main factors. First, the intrinsic size of the antibody and its hydration layer: an IgG is ~150 kDa with an approximate diameter of ~10 nm; once tethered to the nanoparticle surface, the antibody together with its surrounding hydration/ion shell substantially increases the measured hydrodynamic diameter, potentially by over than 10 nm. Secondly, as noted in the revised manuscript, our T-Expand display a high surface density of antibodies; their orientation and conformational extension can enlarge the effective diameter, forming a thicker “soft corona.” While beneficial for functionalization, high conjugation density can thus increase the hydrodynamic size.

To quantify this phenomenon more systematically, we added multimodal measurements. Cryo-TEM analysis of three randomly selected fields yielded a mean T-Expand diameter of 173.1 ± 37.27 nm (Figure 1 h in revised manuscript and added below). Repeat NTA and DLS measurements gave 204 ± 45.8 nm (Figure S11a in revised supplementary materials and inserted below) and 197 ± 39.9 nm (Figure S11b in revised supplementary materials and inserted below), respectively, approximately 24-31 nm larger than cryo-TEM which is consistent with the expected methodological difference that physical diameter (imaging) less than hydrodynamic diameter.

Regarding the “two peaks” distribution observed after antibody conjugation, troubleshooting identified trace contaminants in an earlier NTA microfluidic cartridge; after cleaning and re-measurement, the anomalous subpopulation disappeared. All raw data and images have been included in the revised supplementary materials, source data and are clearly cross-referenced in the response for ease of review.

Figure 1. h, The size distribution of T-Expand, as determined by Cryo-TEM and measured by software of image J with 3 independent figures.

a

b

Supplementary Fig. 11. a, The size distribution of T-Expand, measured by NTA. **b,** The size distribution of T-Expand, measured by DLS.

Point by point reply to Reviewer #3 (Remarks to the Author):

Remarks to the authors:

The manuscript by Zheng et al. presents a well-conceived and technically rigorous study that introduces “T-Expand,” a novel dextran-based nanoparticle platform for T cell activation and CAR T cell manufacturing. The authors demonstrate that T-Expand enables robust T cell expansion, enhanced CD19 CAR T cell cytotoxicity, and in vivo tumor clearance at reduced cell doses. The biocompatibility, high antibody loading capacity, and simplified downstream processing of T-Expand make it a promising alternative to traditional expansion systems such as Dynabeads™. The manuscript is well-structured, the experiments are thorough, and the conclusions are generally well-supported. However, there are several areas where additional data, clarifications, or contextualization would significantly strengthen the claims and translational relevance.

Response: *We thank the reviewer for their generous assessment of our work. We are delighted that you find this study “well-conceived and technically rigorous” and that you recognize the superior performance of the T-Expand platform in T-cell proliferation, CAR T-cell cytotoxicity, and in vivo tumor clearance. We also appreciate your encouragement regarding its biocompatibility, high antibody-loading capacity, and streamlined downstream processing. In the revised manuscript, we will further supplement our data, clarify key details, and expand our Discussion to more fully articulate the advantages and translational potential of T-Expand relative to conventional expansion systems such as Dynabeads™. We have addressed each comment in a point-by-point manner.*

Major issues:

1. The use of azido-dextran and DBCO click chemistry is well-established in immunomaterials literature (e.g., Broaders et al., PNAS, 2009; Adv Mater. 2017;29(7):1603239.). The manuscript should more clearly articulate what is unique about this specific implementation and cite foundational references to distinguish engineering novelty from repurposed techniques.

Response: *Yes, as the reviewer rightly noted, azide–DBCO click chemistry has been extensively applied in immune-materials. We have already cited the references mentioned by the reviewer in our original manuscript (reference 14, 15, 16 in the revised manuscript). In the revised version, we have added detailed text to underscore the unique features of our azide-modified dextran nanoparticle platform and have cited the pertinent literature to delineate its distinctions and innovations compared to existing technologies. Related discussions are provided in line from 455 to 465 in revised manuscript and added below:*

“Compared to existing artificial antigen-presenting nanoparticle carriers, we employed azide-modified dextran polymers to install a high density of azido groups along the side chains for downstream antibody conjugation. In contrast, current polymer platforms,

such as poly(lactic-co-glycolic acid) and poly(ethylene glycol)-block-poly(D,L-lactide), typically localize azido groups at a single terminus of the polymer chain, yielding a low surface functional-group density after nanoparticle assembly. Furthermore, our azide-functionalized dextran nano-platform requires no additional surface modification to undergo click chemistry with DBCO-anti-CD3 and DBCO-anti-CD28, thereby providing an efficient, convenient, and controllable antibody-conjugation system. Finally, these azide-modified dextran nanoparticles remain stable in PBS buffer without loss of azido-group reactivity, offering a ready-to-use antibody-clickable nanocarrier platform”.

2. In the transmission electron microscopy (TEM) image shown in Figure 1g, the T-Expand nanoparticle on the left appears to exhibit a double-layered structure or possibly contains a smaller particle within it. This morphology is not addressed in the text and may lead to confusion regarding particle uniformity. The authors could either clarify this observation in the figure legend or main text, or consider replacing the image with a more representative example that better reflects the described nanoparticle population.

Response: Our T-Expand does not exhibit a bilayer architecture; the apparent bilayer-like features likely arose from impurity particles introduced during sample preparation. To avoid any further confusion, we have replaced them with more representative micrographs (Figure 1 g in revised manuscript) and also provide below.

Figure 1. g, Cryo-TEM visualizing of T-Expand, scale bar: 100 nm.

3. In Figure 5i, the bioluminescence images in row 2, columns 3 and 6 appear to be identical, despite representing different treatment groups. Please verify whether this is a case of inadvertent image duplication.

Response: We thank the reviewer for carefully examining Figure 5i and for identifying this issue. This duplication was entirely our oversight during figure assembly. We have

now corrected the error by replacing the duplicated panel with the appropriate image from the original dataset and have updated the figure accordingly (*Figure 5 i in revised manuscript and added below*). We sincerely appreciate the reviewer's attention to detail in helping us identify and correct this mistake.

Figure 5. i, Representative IVIS overlay images showing bioluminescent signals for each treatment group.

Minor issues:

1. The nanoparticle tracking analysis (NTA) data in Figure 1h suggests a potential bimodal distribution of particle size, with what appears to be two peaks. This raises the question of whether the T-Expand formulation consists of two distinct nanoparticle populations. The authors should elaborate on this in the Results section—either to explain the source of heterogeneity, or to clarify whether this is an artifact of measurement or a true feature of the formulation.

Response: In response to the reviewer's concern regarding the nanoparticle size distribution, we conducted a thorough investigation. First, we confirmed that T-Expand does not comprise multiple particle populations, since it is formed exclusively via strain-promoted azide–alkyne cycloaddition between azide-functionalized dextran nanoparticles and antibodies. Click chemistry is an exceptionally efficient reaction. Therefore, it is unlikely that a fraction of nanoparticles remained unconjugated with antibodies, which could otherwise account for an additional peak population. Two peaks observed in the original NTA data were traced to contamination within the microfluidic sizing channel, which compromised the accuracy of those measurements.

To quantify this phenomenon more systematically, we added multimodal measurements. Cryo-TEM analysis of three randomly selected fields yielded a mean T-Expand diameter of 173.1 ± 37.27 nm (*Figure 1 h in revised manuscript and inserted below*). Repeat NTA and DLS measurements gave 204 ± 45.8 nm (*Figure S11a in revised supplementary materials and added below*) and 197 ± 39.9 nm (*Figure S11b in revised supplementary and inserted below*), respectively, approximately 24–31 nm

larger than cryo-TEM which is consistent with the expected methodological difference that physical diameter (imaging) less than hydrodynamic diameter (solution measurements). Regarding the “two peaks” distribution observed after antibody conjugation, troubleshooting identified trace contaminants in an earlier NTA microfluidic cartridge; after cleaning and re-measurement, the anomalous subpopulation disappeared. All raw data and images have been included in the revised supplementary materials and also inserted below.

Figure 1. h, The size distribution of T-Expand, as determined by Cryo-TEM and measured by software of image J with 3 independent figures.

a

b

Supplementary Fig. 11. a, The size distribution of T-Expand, measured by NTA. **b**, The size distribution of T-Expand, measured by DLS.

2. The representative contour plot in Supplementary Figure 16a appears to display an unusually low number of events, which limits the interpretability of the flow cytometry data. The authors should clarify the number of events collected and justify whether this

sample is truly representative. If possible, consider replacing it with a plot from a higher-quality acquisition to support the conclusions drawn.

Response: We have introduced a new gating strategy of *Figure S19a* in revised supplementary and added below.

Supplementary Fig. 19. T-expand expanded CAR T display antitumor activity in a xenograft model of B-cell lymphoma. (a) Gating strategy of mice spleen and bone marrow on using Flowjo.

Reference

1. Liu, Z., Li, YR., Yang, Y. *et al.* Viscoelastic synthetic antigen-presenting cells for augmenting the potency of cancer therapies. *Nat. Biomed. Eng* **8**, 1615–1633 (2024).
<https://doi.org/10.1038/s41551-024-01272-w>.
2. Casati, A., Varghaei-Nahvi, A., Feldman, S.A. *et al.* Clinical-scale selection and viral transduction of human naïve and central memory CD8+ T cells for adoptive cell therapy of cancer patients. *Cancer Immunol Immunother* **62**, 1563–1573 (2013). <https://doi.org/10.1007/s00262-013-1459-x>
3. https://assets.thermofisher.com/TFS-Assets/LSG/manuals/MAN0008945_SPC0021034_Dynabeads_CD3CD28CTS.pdf
4. Feng, B., Bai, Z., Zhou, X. *et al.* The type 2 cytokine Fc–IL-4 revitalizes exhausted CD8+ T cells against cancer. *Nature* **634**, 712–720 (2024). <https://doi.org/10.1038/s41586-024-07962-4>
5. Kalbasi, A., Siurala, M., Su, L.L. *et al.* Potentiating adoptive cell therapy using synthetic IL-9 receptors. *Nature* **607**, 360–365 (2022). <https://doi.org/10.1038/s41586-022-04801-2>
6. Yamada-Hunter, S.A., Theruvath, J., McIntosh, B.J. *et al.* Engineered CD47 protects T cells for enhanced antitumour immunity. *Nature* **630**, 457–465 (2024). <https://doi.org/10.1038/s41586-024-07443-8>
7. Zhang, A.Q., Hostetler, A., Chen, L.E. *et al.* Universal redirection of CAR T cells against solid tumours via membrane-inserted ligands for the CAR. *Nat. Biomed. Eng* **7**, 1113–1128 (2023).
<https://doi.org/10.1038/s41551-023-01048-8>
8. Zhang, D.K.Y., Adu-Berchie, K., Iyer, S. *et al.* Enhancing CAR-T cell functionality in a patient-specific manner. *Nat Commun* **14**, 506 (2023). <https://doi.org/10.1038/s41467-023-36126-7>
9. Jennifer N. Brudno, James N. Kochenderfer; Toxicities of chimeric antigen receptor T cells: recognition and management. *Blood* 2016; **127** (26): 3321–3330. doi: <https://doi.org/10.1182/blood-2016-04-703751>
10. Norelli, M., Camisa, B., Barbiera, G. *et al.* Monocyte-derived IL-1 and IL-6 are differentially required for cytokine-release syndrome and neurotoxicity due to CAR T cells. *Nat Med* **24**, 739–748 (2018).
<https://doi.org/10.1038/s41591-018-0036-4>.
11. Giavridis, T., van der Stegen, S.J.C., Eyquem, J. *et al.* CAR T cell–induced cytokine release syndrome is mediated by macrophages and abated by IL-1 blockade. *Nat Med* **24**, 731–738 (2018).
<https://doi.org/10.1038/s41591-018-0041-7>.
12. Zhu, E., Yu, J., Li, YR. *et al.* Biomimetic cell stimulation with a graphene oxide antigen-presenting platform for developing T cell-based therapies. *Nat. Nanotechnol.* **19**, 1914–1922 (2024).
<https://doi.org/10.1038/s41565-024-01781-4>
13. Zhang, D.K.Y., Brockman, J.M., Adu-Berchie, K. *et al.* Subcutaneous biodegradable scaffolds for restimulating the antitumour activity of pre-administered CAR-T cells. *Nat. Biomed. Eng* **9**, 268–278 (2025). <https://doi.org/10.1038/s41551-024-01216-4>

Point-by-Point Response to Reviewer's Comments (Zheng et al. formerly manuscript NCOMMS-25-30476A)

For ease of reading, the reviewer's comments are in regular typeface, our responses are highlighted in italics

Point by point reply to Reviewer #2 (Remarks to the Author):

The authors have provided a variety of new/altered data and analysis in response to the original reviews. Unfortunately, however, they did not respond adequately to the main technical concern from the original review - a comparison of the optimized Dynabeads and optimized dextran nanoparticles is required to establish if this new platform provides a true advantage. The experiment here is quite straightforward - repeat previous experiments with a range of Dynabead:cell ratios, similarly to how the studies were performed with their particles, but the authors did not perform.

Response: We thank the reviewer for emphasizing the importance of testing optimized Dynabeads™ conditions. To address this concern, we conducted additional experiments evaluating a range of Dynabeads™:T-cell ratios (0.3:1, 1:1, 3:1, and 5:1), alongside a control group without beads. We have confirmed that a 3:1 Dynabeads™:cell ratio represents the optimized condition for a fair comparison with T-Expand, based on the following factors:

(1) Expansion efficiency:

All Dynabeads™-stimulated groups achieved measurable T-cell expansion by day 10 compared to the unstimulated control. Among the tested conditions, ratios of 3:1 and 5:1 yielded the highest cell numbers (Supplementary Fig. 16a, b in revised supplementary materials and added below), reaching levels consistent with those typically required for clinical-scale CAR T-cell manufacture.

(2) CAR transduction and expansion:

Following lentiviral transduction of CD19 CAR, both the transduction efficiency and expansion fold change were maximal at Dynabeads™:cell ratios of 3:1 and 5:1 (Supplementary Fig. 16 c, d in revised supplementary materials and added below). However, given that the 5:1 condition produced T cells with marginally higher exhaustion marker expression, we identified the 3:1 ratio as the optimal balance between proliferation and phenotypic preservation.

(3) Phenotype analysis:

Flow cytometry profiling revealed that increasing the Dynabeads™:cell ratio above 1:1 resulted, resulted in increased proportion of Terminally differentiated effector memory (TEMRA) T cells accompanied by a gradual decline in the effector-memory (TEM) subset (Supplementary Fig. 16e in

revised supplementary materials and added below). Also, the higher bead-to-cell ratios slightly elevated the expression of exhaustion-associated markers (e.g., PD-1, LAG-3, TIGIT) (*Supplementary Fig. 16f, g, h in revised supplementary materials and added below*).

Collectively, these results confirm that the standard 3:1 Dynabeads™:T-cell ratio represents the optimal condition for robust clinical T-cell activation and CAR T-cell expansion. This finding validates that the comparison in our study that between T-Expand and Dynabeads™ using this optimized (and provide recommended) dose.

We have added the following sentence in the results section: *Optimization experiments revealed that a Dynabeads™:T-cell ratio of 3:1 yielded the most efficient CAR-T cell production (Supplementary Fig. 16a-h). Consequently, this ratio was selected as the benchmark condition for comparison with T-Expand.*

And included the new Supplementary Fig. 16, as shown below.

Supplementary Fig. 16 T cells activated and transduced with CD19 CAR and expanded for 10 days using different Dynabeads™ to cell ratio. (a) Fold expansion of T cell from Day 0 to Day 10. (b) Total absolute cell numbers on Day 10 of expansion (c) Percentage of CAR T cells in total CD3+ T cell population analyzed using flowcytometry (d) Total number of CAR+ T cells on Day 10 (e) Phenotypic distribution of T-cells based on CCR7 and CD45RA expression, showing TN (CD45RA⁺CCR7⁺), TCM (CD45RA⁻CCR7⁺), TEM (CD45RA⁻CCR7⁻), and TEMRA (CD45RA⁺CCR7⁻) subsets. (f-h) T cells from Day 10 stained for exhaustion markers (f) LAG3 (g) TIGIT (h) PD1 and analyzed using flowcytometry.

Point-by-Point Response to Reviewer's Comments (Zheng et al. formerly manuscript NCOMMS-25-30476)

For ease of reading, the reviewer's comments are in regular typeface, our responses are highlighted in italics

Point by point reply to Reviewer #1 (Remarks to the Author):

Reviewer #1 (Remarks to the Author):

The manuscript by Tao Zheng and colleagues presents a comprehensive and innovative study describing the development and evaluation of a novel dextran-based nanoparticle platform, termed T-Expand, for the ex vivo expansion of CAR T cells. According to the authors, this platform addresses key limitations of current artificial antigen-presenting cell (aAPC) systems such as Dynabeads™ and TransAct™, including overstimulation, poor scalability, and challenges in manufacturing.

In the first part of the study, the authors design and characterize the T-Expand nanoparticles, which feature a high surface density of anti-CD3 and anti-CD28 antibodies conjugated via click chemistry to the modifiable side chains of dextran. They claim that the nanoscale size and biocompatibility of T-Expand eliminate the need for post-expansion removal, simplifying the manufacturing process and improving scalability.

In the second part of the study, the functional characteristics of CAR T cells expanded using T-Expand are assessed. The results suggest that these cells possess enhanced phenotypic and functional attributes, including a greater proportion of CD27⁺ and CD28⁺ effector memory T cells, reduced expression of exhaustion markers, and improved anti-tumor efficacy both in vitro and in vivo. Furthermore, bulk RNA sequencing reveals upregulation of effector genes, including cytokines, and chemokine receptors implicated in T cell trafficking and immune modulation. The authors propose that these transcriptional changes correlate with the superior in vivo persistence and anti-tumor performance observed in xenograft lymphoma models, where T-Expand-expanded CAR T cells outperformed higher-dose Dynabeads™-expanded cells.

This is a timely and overall well-executed study that introduces a promising new platform for CAR T-cell manufacturing. It represents a meaningful contribution to the field of adoptive cell therapy, with the potential to improve therapeutic efficacy and streamline production. However, before publication, the authors should address the following major and minor concerns:

Major concerns

1. The authors report the optimization of anti-CD3 and anti-CD28 antibody ratios used for the functionalization of the T-Expand particles. However, no data are provided regarding the final amount of each antibody successfully conjugated to the particles. Since conjugation efficiency may differ from the input ratios, quantification (e.g., using ELISA or fluorophore-tagged antibodies) is important to clearly report the final antibody composition present on the T-Expand particles used in the experiments.

*Response: In the revised manuscript, we have incorporated a two-step fluorometric analysis to investigate the amount of conjugated antibody density on the surface of NP, now detailed in the **Results** sections and **Methods** (Figure 1f in revised manuscript, Supplementary Fig. 10 in revised supplementary materials), with all raw data provided in the **source data**. Details of the added data, is described below:*

Step A – Stoichiometric verification.

To confirm that the surface ratio of anti-CD3 to anti-CD28 matches the input ratio, anti-CD3 was labeled with Cy5-NHS and anti-CD28 with Alexa Fluor 488-NHS; both antibodies were simultaneously functionalized with DBCO-Sulfo-NHS (Figure S1, a). A microplate reader was used to quantify the Anti-CD3 and Anti-CD28 fluorescence calibration curves (Figure S1, b). Then, DBCO-Cy5-anti-CD3 and DBCO-Alexa Fluor 488-anti-CD28 at different concentration ratios (3:1; 1:1; 1:3) were subjected to click coupling reaction with naked NP. After the click-coupling reaction, NPs were pelleted by centrifugation. Fluorescence intensities of the supernatants were quantified against calibration curves for each antibody, allowing us to back-calculate the amount of antibody bound. The bounded anti-CD3:anti-CD28 ratio on the particle surface matched the input ratio within experimental error (Supplementary Fig. 10a, b in revised supplementary materials and added below).

Step B – Absolute loading determination.

To obtain a more accurate estimate of the amount of antibody on the nanoparticle surface, we first confirmed that the CD3:CD28 ratio on the particles matched the input ratio, because concurrent labeling of an antibody with both a fluorophore and DBCO-Sulfo-NHS can introduce competition between the two reagents and potentially underrepresent the total amount conjugated. We next adopted a single bifunctional tag. Specifically, anti-CD28 was derivatized with Cy5-DBCO-NHS, eliminating any competition and enabling direct fluorescence quantification. A calibration curve for Cy5-DBCO-anti-CD28 was generated with a microplate reader, and the resulting equation was applied to the supernatant measurements to calculate the mass of antibody bound per milligram of nanoparticles (Supplementary Fig. 10c, d in revised supplementary materials and added below). The final loading was $18.2 \pm 3.1 \mu\text{g}$ of

antibody per milligram of dextran nanoparticles (*Figure 1f in revised manuscript and added below*).

Together, these experiments demonstrate (i) that the final anti-CD3/anti-CD28 stoichiometry on T-Expand is controlled by the feed ratio, and (ii) that the platform attains an average antibody amount of $\sim 18 \mu\text{g mg}^{-1}$ NP. We believe these additions substantially strengthen the rigor and reproducibility of the study, and we are grateful for the reviewer's constructive recommendation. *The related description was provided in the result section, line 137 to 147 in revised manuscript.*

Supplementary Fig. 10. Quantification of conjugated antibody density on the surface of NP. The antibody conjugation efficiency on the surface of naked NPs measured by Nanodrop. **a**, Quantification of the conjugated anti-CD3 and anti-CD28 antibodies ratio on the surface of dextran NPs using a fluorescence spectroscopy microplate reader. **b**, anti-CD3 calibration curve used for final quantification and anti-CD28 calibration curve used for final quantification. **c**, Quantification of the conjugated anti-CD28 antibodies on the surface of dextran NPs using a fluorescence spectroscopy microplate reader. **d**, anti-CD28 calibration curve used for final quantification.

Figure 1f. The antibody conjugation efficiency on the surface of naked NPs measured by microplate reader.

2. While the authors emphasize the biocompatibility of the T-Expand nanoparticles as a key feature, no clear experimental data are provided to support this claim. The presented T cell proliferation data do not address the concentration-dependent effects of the particles on the overall cell population. In this context, I recommend including a biocompatibility assessment, such as cytotoxicity or cell proliferation assays using T-Expand particles at varying concentrations on T cells and/or other relevant cell lines, to confirm the system's safety profile.

Response: To verify the biocompatibility of our platform, we have added a dedicated biocompatibility study to the revised manuscript. Jurkat T cells were co-incubated with a broad range of either naked dextran nanoparticles or T-Expand and seeded into 96-well plates at 5×10^5 cells per well. Cell viability was assessed with the NucleoCounter NC-200 on days 1, 4, 7, and 10, following the manufacturer's protocol. Across all concentrations and time points, cell survival consistently exceeded 90%, demonstrating excellent biocompatibility for both naked NPs and T-Expand. The complete experimental protocol has been added as a new "Biocompatibility Assay of T-Expand" subsection in the **Methods** (line 696 to 701 in revised manuscript.) and related discussion has been inserted in the line of 162 to 164 in the revised manuscript. the quantitative data and statistical analyses are presented in the Results section (Supplementary Fig. 14 in revised supplementary materials and added below).

Supplementary Fig. 14. Quantification of cell viability (determined by NucleoCounter NC-200 machine) at day 1, 4, 7 and 10 by co-culturing naked NPs and T-Expand with Jurkat T cells.

3. Did the authors use equivalent concentrations of Dynabeads™, TransAct™, and T-Expand particles, considering their differing physical and biochemical nature? Specifically, do these platforms contain comparable amounts of anti-CD3 and anti-CD28 antibodies? Given that T-Expand is a nanoparticle-based system, while Dynabeads™ and TransAct™ are microscale or polymer-based platforms, direct comparison may be confounded by differences in particle size, surface area, and antibody density. For example, was normalization performed based on the number of particles, the total antibody content, surface density of antibodies, or functional activity? The authors should clearly describe and justify the approach used to compare the different platforms and discuss whether the different concentrations used may lead to different results.

Response: We thank the reviewer for highlighting the critical issue of dose normalization. Because the total antibody load, CD3:CD28 stoichiometry, and conjugation chemistry for Dynabeads™ and TransAct™ are proprietary, only limited information is publicly available, that Dynabeads™ display around 2.8×10^5 antibody molecules per bead with an anti-CD3:anti-CD28 molar ratio of ~3:1 [1], whereas

TransAct™ is reported to present a 1:1 ratio but without a disclosed antibody amount [2]. These commercial microscale carriers therefore differ markedly from our nanoscale T-Expand particles in diameter, specific surface area, and ligand density, making it challenging to normalize on the basis of absolute antibody site number.

We therefore adopted a clinically oriented comparison, The T-Expand and Dynabeads™ were seeded with the same initial number of T cells; the dose of Dynabeads™ (beads to cell ratio = 3:1) were used exactly according to commercial (thermo fisher scientific) recommendation for clinical CAR T cell manufacture protocol [3] and adopted by many research group [4-8] while T-Expand was tested across a 0.75µg to 30 µg dose of range T-Expand to identify the optimal concentration. The unified read-out was the fold expansion achieved after 10 days. The dose–response data show that T-Expand at ≥15 µg attains a cell-expansion efficiency comparable to Dynabeads™ used at a 3:1 bead-to-cell ratio (Figure 3c in revised manuscript, Figure S16 in revised supplementary materials). This design ensures the comparator was tested at its recommended clinical baseline while allowing a fair search for T-Expand's optimum.

4. The authors claim that the particles do not require post-expansion removal due to their nanoscale size and biocompatibility. However, this assumes the particles remain stable and do not aggregate, accumulate, or degrade over time. I strongly recommend assessing the stability and integrity of T-Expand particles over relevant time points under physiological experimental conditions.

Response: *Thank you for your attention to our statement that “no removal is required.” By this, we mean that T-Expand, due to their nanoscale dimensions, are effectively eliminated during standard T-cell washing procedures, obviating the magnetic-separation step necessary for Dynabeads™. In support of this, We conducted an additional experiment in which T-Expand was incubated with T cells for varying durations, followed by centrifugal washing. TEM imaging of the washed T cells was then performed to assess whether T-Expand remained on the cell surface or in the suspension, thereby demonstrating that T-Expand can be effectively removed by standard washing procedures. The TEM imaging confirmed the absence of T-Expand on the T-cell surface or in the post-wash suspension (Figure S15 in revised supplementary materials and inserted below). To further substantiate this attribute and assess nanoparticle stability under physiologically relevant conditions, we have incorporated the following two experimental series.*

First, supernatants were collected at various time points following T-cell expansion with either T-Expand (15 µg/mL) or Dynabeads™ (75 µL/mL, 3:1 beads to cell ratio). These supernatants were then diluted 1:1 and applied to fresh, unactivated primary T cells, and activation was measured 20 hours later by CD69/CD25 co-expression. The stimulatory potency of the T-Expand supernatant declined progressively after day 6 (Figure 1i in revised manuscript and inserted below). To eliminate any cell-mediated contributions, we incubated T-Expand and Dynabeads™ in cell-free RPMI 1640 (10%

FBS, 37 °C, 5% CO₂), collected the media on days 2, 6, and 10, and used them directly to stimulate primary T cells. Under these acellular conditions, T-Expand lost significant activity by day 6, whereas Dynabeads™ retained a relatively stable stimulatory effect (Figure 1j in revised manuscript and inserted below). These data indicate that the active moieties of T-Expand are gradually lost over time. This means, that in a clinical use scenario, even residual, inadequately washed T-Expand fails to activate T cells. The experimental design, results, and corresponding discussion have been added to the revised manuscript to address the concerns related to stability, and need for removal of T-Expand. The related discussion has been added in the line of 165 to 179 in the revised manuscript.

Supplementary Fig. 15. Representative transmission electron microscopy (TEM) images of primary T cell and activated at different time point (15 mins, 30 mins, 120 mins and 24 h). Scale bars: 5 μ m.

i

j

Figure 1. *i*, the T cell activation profile in terms of % of T cells expressing CD69+CD25+ after 20 hrs of co-incubation with the supernatant obtained from the expansion cultures at different time point. *j*, The T-Expand (20 μ l) and Dynabeads™ (75 μ l) were incubated in Roswell Park Memorial Institute (RPMI) 10%FCS at 37°C and 5% CO₂ culture conditions for 10 days and the media was removed at day 2,6 and 10 timepoints of culture and was added to unactivated T cells at 1:1 dilution. Data in panels *i*, *j* represented as mean \pm SD, of biological replicates with each dot indicating one healthy donor sample($n=1$).

5. RNA-seq data indicate substantial upregulation of IFN- γ , pro-inflammatory cytokines (e.g., IL-6, IL-17), and other immune mediators in T-Expand-expanded CAR T cells. While these signatures may correlate with increased functionality, they also raise concerns about the potential for cytokine release syndrome (CRS) and immune effector cell-associated neurotoxicity syndrome (ICANS), two major clinical risks associated with CAR T-cell therapy. The authors should explicitly discuss these risks in the manuscript and consider whether the upregulated cytokine profiles observed for the T-Expand approach may pose translational safety concerns. In this context, it would be informative if the authors could report and discuss whether they observed any adverse reactions in their CAR T-cell experiments in mice.

Response: We thank the reviewer for this important and clinically relevant comment. We agree that the upregulation of IFN- γ and pro-inflammatory cytokines, such as IL-6 and IL-17, observed in our RNA-seq analysis could, in principle, be associated with cytokine release syndrome (CRS) or immune effector cell-associated neurotoxicity syndrome (ICANS) in the clinical setting. In the revised Discussion section, we now explicitly address these potential translational safety concerns. Related discussions are provided *in line from 540 to 553 in revised manuscript and added below:*

“There is marked upregulation of pro-inflammatory cytokine transcripts (IFN- γ , IL-6, IL-17) in T-Expand–expanded CAR T cells relative to Dynabeads™. While such signatures are consistent with heightened effector programming, they overlap with pathways implicated in clinical cytokine release syndrome (CRS) and immune-effector cell–associated neurotoxicity syndrome (ICANS) [9]. Importantly, bulk transcriptomic elevation does not equate to systemic cytokine release, and in patients the surge in IL-6 during CRS is predominantly mediated by monocyte/macrophage activation rather than by T cells themselves [10, 11].

In our NSG mouse xenograft experiments, we did not observe any adverse reactions, including distress, weight loss, behavioral changes, during the treatment period, despite the potent antitumor activity of T-Expand–expanded CAR T cells. Considering that NSG mice are not an appropriate system to model CRS and ICANS, this should be evaluated in future studies using immune-competent and humanized animal models in which myeloid-driven cytokine biology is preserved”.

6. Additional discussion is needed regarding the differing functional outcomes observed between the Dynabeads™ and T-Expand platforms. This is particularly important, as the two systems yield distinct clinical-relevant T cell activities and gene expression profiles. How do the authors explain the observed divergence in activity? Is the unique nature of the T-Expand nanoparticles, such as their size, flexibility, or nanoscale interaction dynamics, sufficient to account for these changes? Could the chemical composition of the dextran polymer itself contribute to altered T cell activation or downstream signaling? Alternatively, might these differences arise from distinct physicochemical properties of the particles, such as surface charge, stiffness, or antibody orientation and density? The manuscript currently lacks a detailed analysis and discussion regarding this point and a thoughtful discussion of these mechanisms would greatly strengthen the manuscript and support the claims made about the advantages of the T-Expand system over conventional aAPC platforms.

Response: *Based on the activation assay, we confirm that neither naked dextran nanoparticles nor the azide-modified polymer backbone activate T cells or alter downstream signaling (Figure 1e in the revised manuscript and inserted below). We also show that T-Expand and Dynabeads™ have similar surface zeta potentials (Figure 1d in the revised manuscript and inserted below), and although Atomic Force Microscopy (AFM) imaging measurements reveal that T-Expand’s surface Young’s modulus (488 Kpa, Figure 1 and table 1 shown below) is much lower than Dynabeads™ (20–40 MPa), both platforms still effectively expand T cells, implying stiffness is not a decisive factor here. Although we lack orientation data for Dynabeads™ or TransAct™, the fact that T-Expand retains high stimulatory potency after serum incubation (Figure S12,13 in revised supplementary materials) indicates that antibody orientation is not limiting on our platform. Related discussions are provided in line from 489 to 506 in revised manuscript and added below:*

“Mechanistically, the divergent functional outcomes of T-Expand and Dynabeads™ appear to be driven by differences in stimulus intensity and kinetics. T-Expand presents a higher surface density of anti-CD3/anti-CD28, delivering more activation signals and enabling the rapid formation of multiple immunological synapses. This brief, high-density “burst” is sufficient to initiate T-cell activation and proliferation without prolonged high-strength engagement. By contrast, we observed that Dynabeads™ remain tightly connected to T-cell membranes throughout the expansion period, providing continuous strong signaling that while promoting early proliferation, may also predispose cells to activation-induced exhaustion. In addition, the size and biocompatibility of T-expand facilitates receptor-mediated internalization, which aligns with the natural T cell behavior upon TCR activation. We hypothesize that the internalization capacity contributes to the observed phenotype.

Consistent with this model, our stimulation-stability assays show that, under cell-free conditions, T-Expand’s stimulatory activity declines by day 6, indicating natural dissipation/degradation that limits prolonged overstimulation, whereas Dynabeads™ maintain relatively constant potency. In line with this, CAR-T cells expanded with Dynabeads™ exhibit higher exhaustion-marker expression in a three-round in vitro tumor rechallenge assay, whereas those expanded with T-Expand retain a less-exhausted CD27⁺CD28⁺ phenotype and superior durability under repeated challenge”.

Figure 1. d, Zeta potential of naked NPs, T-Expand and Dynabeads™ (n=3). **e**, Percent of T cells expressing the activation markers CD25 and CD69 after co-incubation with T-Expand (number of donors: n = 3).

Figure 1. AFM topography image of the particles.

Table 1. Nanomechanical properties measured by AFM

Sample	Young Modulus (KPa)
T-Expand	488
Dynabeads™	20000-40000 [1]

Other comments:

1. In lines 142–143, the authors state that T-Expand particles exhibit high uniformity in cryo-EM images. However, no quantitative analysis is presented from the cryoEM micrographs. Instead, the authors rely on NTA data showing a mean particle size of 199.8 ± 70.7 nm. Figure 1h shows a secondary peak at ~ 120 nm, suggesting the presence of a distinct subpopulation. The authors should discuss this heterogeneity and clarify whether the smaller peak represents unbound particles. If possible, quantify and report the proportion of each population.

Response: In accordance with the suggestion, we re-imaged T-Expand by cryo-TEM. The resulting mean diameters for T-Expand were 173.1 ± 37.27 nm as quantify by image J. These data, now presented as histograms with population percentages in figure 1h in revised manuscript and inserted below.

Figure 1. h, The size distribution of T-Expand, as determined by Cryo-TEM and measured by software of image J with 3 independent figures.

Additionally, we do not attribute peak of 120 nm to unconjugated particles. Normally, the strain-promoted alkyne–azide cycloaddition (SPAAC) between azide-decorated NPs and DBCO-antibodies proceeds with high efficiency, and NanoDrop spectroscopy at 309 nm confirms successful incorporation of DBCO into the antibody preparation (Figure S9 in revised supplementary materials). Owing to the high efficiency of the strain-promoted alkyne–azide cycloaddition between azide-functionalized nanoparticles and DBCO-modified antibodies, the ~120 nm peak does not represent nanoparticle populations lacking antibody conjugation.

Finally, we repeated the NTA measurements and investigated the previously observed subpopulation peak. We identified minor contaminants in the original NTA microfluidic chip. After thorough cleaning of the chip and re-analysis, the subpopulation peak was no longer detected (Figure S11a in revised supplementary materials and added below). Concurrently, we performed DLS-based particle sizing as a complementary control (Figure S11b in revised supplementary materials and inserted below), which yielded an average T-Expand particle diameter of approximately 200 nm and consistent with the NTA results. The size measured by NTA and DLS is large than Cryo-TEM imaging, this is because the physical diameter of the NP (as determined by TEM imaging) is typically smaller than its hydrodynamic diameter.

a

b

Supplementary Fig. 11. a, The size distribution of T-Expand, measured by NTA. **b**, The size distribution of T-Expand, measured by DLS.

2. Overall, the authors should provide more comprehensive descriptions of their experimental procedures, as the current level of detail is insufficient for reproducibility. For example, in lines 501–514, the section describing the formation of T-Expand particles for T cell activation and expansion lacks critical experimental information. This includes the concentrations of antibodies used, reaction volumes, buffer compositions, solvents, and technical details of the instrumentation employed. The authors are encouraged to thoroughly revise and complete all experimental sections. Specifically, the sources and references for the anti-CD3 and anti-CD28 antibodies should be clearly stated. Additionally, the origin of all cell lines used, along with their corresponding unique identifiers or commercial catalog numbers, should be provided in the methods to ensure traceability.

Response: We thank the reviewer for raising this point. We have revised the **Methods** section throughout to ensure that details are provided. Specifically, we have expanded the description of T-Expand particle preparation and T-cell activation to include all

concentrations, reaction volumes, buffer compositions, solvents, and relevant instrument specifications. We have also provided the sources, catalog numbers, and unique identifiers for anti-CD3 and anti-CD28 antibodies, as well as the origin and catalog information for all cell lines used. These additions ensure full traceability and reproducibility of our experiments.

Point by point reply to Reviewer #2 (Remarks to the Author):

This manuscript describes dextran nanoparticles to which T cell stimulating antibodies have been conjugated using click chemistry, and their use to promote T cell and CAR-T cell expansion, along with analysis of the function of the resultant cells. Overall, the authors provide a set of very nice studies characterizing the nanoparticles, and T cell expansion with these dextran-based nanoparticles. One major concern is the novelty of this work. A number of nanoparticle and microparticle artificial antigen presenting cells (aAPCs) have been developed and published, so the novelty of the concept is not high. The authors highlight the improvements, as versus Dynabeads, that they find when using these particles to expand T cells, particularly in relation to the functionality of the resultant cell populations. However, it is not clear the authors provide a fair comparison, as they only compare to a single Dynabead condition. The authors first optimize the dose of their nanoparticles to obtain optimal expansion and T cell phenotype, and then utilize that optimal dose for subsequent studies. The finding that stimulation dose is critical to expansion and function of the T cells is consistent with past literature using other aAPCs. So a different, optimized, dose of Dynabeads in these present studies may result in very similar, beneficial impact on CAR-T expansion, differentiation status, and function. Without a comparison of the optimized Dynabeads and optimized dextran nanoparticles, one cannot know if one provides a true advantage.

Technical question – the size distribution of the dextran nanoparticles have a much larger size after antibody conjugation than prior. How do the authors explain? Also, the size distribution post-antibody conjugation has two peaks; one approx. the same size as reported for the naked NPs, the other much larger – how do the authors account for this

Response: *We thank the reviewer for their positive comments related to our work. We have addressed each comment in a point-by-point manner.*

*First of all, based on the novelty concerns, while various nanoparticle- and microparticle-based aAPC platforms have been reported, T-Expand offers three key innovations that substantially enhance its novelty and translational potential (shown below). **Related discussions are provided in line from 455 to 465 in revised manuscript.***

1. High-Density Ligand Loading.

It is well established that optimal T-cell activation and proliferation depend on parameters such as particle size, shape, ligand density, and rigidity. Micron-scale particles typically excel at T-cell stimulation because their larger contact area promotes more immunological synapse formation and greater ligand–receptor pairing. However, to achieve comparable activation efficiency at the nanoscale, aAPCs must present very high ligand densities within their limited contact footprint. Existing polymer-based systems, such as PLGA and PEG-PDLLA nanoparticles, only introduce functional groups at chain termini, resulting in sparse surface click-handles after self-assembly.

In contrast, our azide-functionalized dextran features dense azido groups along every side chain, dramatically increasing antibody loading per particle. This high ligand density enables powerful T-cell activation and expansion at the nanoscale. Moreover, we have carried out extensive functional studies of CD19 CAR-T cells expanded with T-Expand, demonstrating enhanced cytotoxicity, persistence, and phenotype both in vitro and in vivo.

2. One-Step Bioorthogonal Conjugation.

Most reported aAPC platforms ranging from inorganic beads to polymeric microparticles which require multiple surface modification steps and secondary linkers to attach stimulatory antibodies [1, 12, 13]. By contrast, azide-modified dextran nanoparticles react directly with DBCO-anti-CD3 and DBCO-anti-CD28 in a single strain-promoted alkyne-azide cycloaddition (“click”) step. This one-pot process eliminates the need for additional activation or coupling chemistry, greatly simplifying manufacturing and reducing batch variability.

Figure 1. Reference of related aAPCs. **a.** alginate microgel based aAPCs. **b.** graphite micro-sheets based aAPCs. **c.** Silica microrod based aAPCs.

3. Beyond T-Cell Stimulation: Impact of T-Expand on CAR T-Cell Expansion and Function

In addition to the general advances of nanoscale aAPCs, we would like to emphasize that our study goes beyond T-cell stimulation alone. Specifically, we extend the application of the T-Expand platform into the clinically critical context of CAR T-cell manufacturing. We provide an in-depth analysis of how T-Expand-expanded CAR T cells differ in phenotype, persistence, and anti-tumor efficacy. To our knowledge, very few reports of nanoscale aAPCs have systematically investigated their impact on CAR T-cell expansion and downstream therapeutic performance. This extension into CAR T-cell production and functional evaluation further distinguishes our work and underscores its translational relevance.

Without a comparison of the optimized Dynabeads and optimized dextran nanoparticles, one cannot know if one provides a true advantage.

Response: *We thank the reviewer for highlighting the optimized dose comparison. Because the total antibody load, CD3:CD28 stoichiometry, and conjugation chemistry for Dynabeads™ and TransAct™ are proprietary, only limited information is publicly available. According to the reported work that Dynabeads™ display $\approx 2.8 \times 10^5$ antibody molecules per bead with an anti-CD3:anti-CD28 molar ratio of $\sim 3:1$ [1], whereas TransAct™ is reported to present a 1:1 ratio but without a disclosed antibody amount [2]. These commercial microscale carriers therefore differ markedly from our nanoscale T-Expand particles in diameter, specific surface area, and ligand density, making it challenging to normalize on the basis of absolute antibody site number.*

We therefore adopted a clinically oriented comparison, The T-Expand and Dynabeads™ were seeded with the same initial number of T cells; the dose of Dynabeads™ (beads to cell ratio = 3:1) were used exactly according to commercial (thermo fisher scientific) recommendation for clinical CAR T cell manufacture protocol [3] and adopted by many research works [4-8] while T-Expand was tested across a 0.75 μg –30 μg dose of range T-Expand to identify the optimal concentration. The unified read-out was the fold expansion achieved after 10 days. The data of dose–response data show that T-Expand at $\geq 15 \mu\text{g}$ attains a cell-expansion efficiency comparable to Dynabeads™ used at a 3:1 bead-to-cell ratio (Figure 3c in revised manuscript, Figure S16 in revised supplementary materials). This design ensures the comparator was tested at its recommended clinical baseline while allowing a fair search for T-Expand’s optimum.

Technical question – the size distribution of the dextran nanoparticles have a much larger size after antibody conjugation than prior. How do the authors explain?

Response: We attribute the apparent increase in particle size after antibody conjugation to two main factors. First, the intrinsic size of the antibody and its hydration layer: an IgG is ~150 kDa with an approximate diameter of ~10 nm; once tethered to the nanoparticle surface, the antibody together with its surrounding hydration/ion shell substantially increases the measured hydrodynamic diameter, potentially by over than 10 nm. Secondly, as noted in the revised manuscript, our T-Expand display a high surface density of antibodies; their orientation and conformational extension can enlarge the effective diameter, forming a thicker “soft corona.” While beneficial for functionalization, high conjugation density can thus increase the hydrodynamic size.

To quantify this phenomenon more systematically, we added multimodal measurements. Cryo-TEM analysis of three randomly selected fields yielded a mean T-Expand diameter of 173.1 ± 37.27 nm (Figure 1 h in revised manuscript and added below). Repeat NTA and DLS measurements gave 204 ± 45.8 nm (Figure S11a in revised supplementary materials and inserted below) and 197 ± 39.9 nm (Figure S11b in revised supplementary materials and inserted below), respectively, approximately 24-31 nm larger than cryo-TEM which is consistent with the expected methodological difference that physical diameter (imaging) less than hydrodynamic diameter.

Regarding the “two peaks” distribution observed after antibody conjugation, troubleshooting identified trace contaminants in an earlier NTA microfluidic cartridge; after cleaning and re-measurement, the anomalous subpopulation disappeared. All raw data and images have been included in the revised supplementary materials, source data and are clearly cross-referenced in the response for ease of review.

Figure 1. h, The size distribution of T-Expand, as determined by Cryo-TEM and measured by software of image J with 3 independent figures.

a

b

Supplementary Fig. 11. a, The size distribution of T-Expand, measured by NTA. **b,** The size distribution of T-Expand, measured by DLS.

Point by point reply to Reviewer #3 (Remarks to the Author):

Remarks to the authors:

The manuscript by Zheng et al. presents a well-conceived and technically rigorous study that introduces “T-Expand,” a novel dextran-based nanoparticle platform for T cell activation and CAR T cell manufacturing. The authors demonstrate that T-Expand enables robust T cell expansion, enhanced CD19 CAR T cell cytotoxicity, and in vivo tumor clearance at reduced cell doses. The biocompatibility, high antibody loading capacity, and simplified downstream processing of T-Expand make it a promising alternative to traditional expansion systems such as Dynabeads™. The manuscript is well-structured, the experiments are thorough, and the conclusions are generally well-supported. However, there are several areas where additional data, clarifications, or contextualization would significantly strengthen the claims and translational relevance.

Response: *We thank the reviewer for their generous assessment of our work. We are delighted that you find this study “well-conceived and technically rigorous” and that you recognize the superior performance of the T-Expand platform in T-cell proliferation, CAR T-cell cytotoxicity, and in vivo tumor clearance. We also appreciate your encouragement regarding its biocompatibility, high antibody-loading capacity, and streamlined downstream processing. In the revised manuscript, we will further supplement our data, clarify key details, and expand our Discussion to more fully articulate the advantages and translational potential of T-Expand relative to conventional expansion systems such as Dynabeads™. We have addressed each comment in a point-by-point manner.*

Major issues:

1. The use of azido-dextran and DBCO click chemistry is well-established in immunomaterials literature (e.g., Broaders et al., PNAS, 2009; Adv Mater. 2017;29(7):1603239.). The manuscript should more clearly articulate what is unique about this specific implementation and cite foundational references to distinguish engineering novelty from repurposed techniques.

Response: *Yes, as the reviewer rightly noted, azide–DBCO click chemistry has been extensively applied in immune-materials. We have already cited the references mentioned by the reviewer in our original manuscript (reference 14, 15, 16 in the revised manuscript). In the revised version, we have added detailed text to underscore the unique features of our azide-modified dextran nanoparticle platform and have cited the pertinent literature to delineate its distinctions and innovations compared to existing technologies. Related discussions are provided in line from 455 to 465 in revised manuscript and added below:*

“Compared to existing artificial antigen-presenting nanoparticle carriers, we employed azide-modified dextran polymers to install a high density of azido groups along the side chains for downstream antibody conjugation. In contrast, current polymer platforms,

such as poly(lactic-co-glycolic acid) and poly(ethylene glycol)-block-poly(D,L-lactide), typically localize azido groups at a single terminus of the polymer chain, yielding a low surface functional-group density after nanoparticle assembly. Furthermore, our azide-functionalized dextran nano-platform requires no additional surface modification to undergo click chemistry with DBCO-anti-CD3 and DBCO-anti-CD28, thereby providing an efficient, convenient, and controllable antibody-conjugation system. Finally, these azide-modified dextran nanoparticles remain stable in PBS buffer without loss of azido-group reactivity, offering a ready-to-use antibody-clickable nanocarrier platform”.

2. In the transmission electron microscopy (TEM) image shown in Figure 1g, the T-Expand nanoparticle on the left appears to exhibit a double-layered structure or possibly contains a smaller particle within it. This morphology is not addressed in the text and may lead to confusion regarding particle uniformity. The authors could either clarify this observation in the figure legend or main text, or consider replacing the image with a more representative example that better reflects the described nanoparticle population.

Response: Our T-Expand does not exhibit a bilayer architecture; the apparent bilayer-like features likely arose from impurity particles introduced during sample preparation. To avoid any further confusion, we have replaced them with more representative micrographs (Figure 1g in revised manuscript) and also provide below.

Figure 1. g, Cryo-TEM visualizing of T-Expand, scale bar: 100 nm.

3. In Figure 5i, the bioluminescence images in row 2, columns 3 and 6 appear to be identical, despite representing different treatment groups. Please verify whether this is a case of inadvertent image duplication.

Response: We thank the reviewer for carefully examining Figure 5i and for identifying this issue. This duplication was entirely our oversight during figure assembly. We have

now corrected the error by replacing the duplicated panel with the appropriate image from the original dataset and have updated the figure accordingly (*Figure 5 i in revised manuscript and added below*). We sincerely appreciate the reviewer's attention to detail in helping us identify and correct this mistake.

Figure 5. i, Representative IVIS overlay images showing bioluminescent signals for each treatment group.

Minor issues:

1. The nanoparticle tracking analysis (NTA) data in Figure 1h suggests a potential bimodal distribution of particle size, with what appears to be two peaks. This raises the question of whether the T-Expand formulation consists of two distinct nanoparticle populations. The authors should elaborate on this in the Results section—either to explain the source of heterogeneity, or to clarify whether this is an artifact of measurement or a true feature of the formulation.

Response: In response to the reviewer's concern regarding the nanoparticle size distribution, we conducted a thorough investigation. First, we confirmed that T-Expand does not comprise multiple particle populations, since it is formed exclusively via strain-promoted azide–alkyne cycloaddition between azide-functionalized dextran nanoparticles and antibodies. Click chemistry is an exceptionally efficient reaction. Therefore, it is unlikely that a fraction of nanoparticles remained unconjugated with antibodies, which could otherwise account for an additional peak population. Two peaks observed in the original NTA data were traced to contamination within the microfluidic sizing channel, which compromised the accuracy of those measurements.

To quantify this phenomenon more systematically, we added multimodal measurements. Cryo-TEM analysis of three randomly selected fields yielded a mean T-Expand diameter of 173.1 ± 37.27 nm (*Figure 1 h in revised manuscript and inserted below*). Repeat NTA and DLS measurements gave 204 ± 45.8 nm (*Figure S11a in revised supplementary materials and added below*) and 197 ± 39.9 nm (*Figure S11b in revised supplementary and inserted below*), respectively, approximately 24–31 nm

larger than cryo-TEM which is consistent with the expected methodological difference that physical diameter (imaging) less than hydrodynamic diameter (solution measurements). Regarding the “two peaks” distribution observed after antibody conjugation, troubleshooting identified trace contaminants in an earlier NTA microfluidic cartridge; after cleaning and re-measurement, the anomalous subpopulation disappeared. All raw data and images have been included in the revised supplementary materials and also inserted below.

Figure 1. h, The size distribution of T-Expand, as determined by Cryo-TEM and measured by software of image J with 3 independent figures.

a

b

Supplementary Fig. 11. a, The size distribution of T-Expand, measured by NTA. **b**, The size distribution of T-Expand, measured by DLS.

2. The representative contour plot in Supplementary Figure 16a appears to display an unusually low number of events, which limits the interpretability of the flow cytometry data. The authors should clarify the number of events collected and justify whether this

sample is truly representative. If possible, consider replacing it with a plot from a higher-quality acquisition to support the conclusions drawn.

Response: We have introduced a new gating strategy of *Figure S19a* in revised supplementary and added below.

Supplementary Fig. 19. T-expand expanded CAR T display antitumor activity in a xenograft model of B-cell lymphoma. (a) Gating strategy of mice spleen and bone marrow on using Flowjo.

Reference

1. Liu, Z., Li, YR., Yang, Y. *et al.* Viscoelastic synthetic antigen-presenting cells for augmenting the potency of cancer therapies. *Nat. Biomed. Eng* **8**, 1615–1633 (2024).
<https://doi.org/10.1038/s41551-024-01272-w>.
2. Casati, A., Varghaei-Nahvi, A., Feldman, S.A. *et al.* Clinical-scale selection and viral transduction of human naïve and central memory CD8+ T cells for adoptive cell therapy of cancer patients. *Cancer Immunol Immunother* **62**, 1563–1573 (2013). <https://doi.org/10.1007/s00262-013-1459-x>
3. https://assets.thermofisher.com/TFS-Assets/LSG/manuals/MAN0008945_SPC0021034_Dynabeads_CD3CD28CTS.pdf
4. Feng, B., Bai, Z., Zhou, X. *et al.* The type 2 cytokine Fc–IL-4 revitalizes exhausted CD8+ T cells against cancer. *Nature* **634**, 712–720 (2024). <https://doi.org/10.1038/s41586-024-07962-4>
5. Kalbasi, A., Siurala, M., Su, L.L. *et al.* Potentiating adoptive cell therapy using synthetic IL-9 receptors. *Nature* **607**, 360–365 (2022). <https://doi.org/10.1038/s41586-022-04801-2>
6. Yamada-Hunter, S.A., Theruvath, J., McIntosh, B.J. *et al.* Engineered CD47 protects T cells for enhanced antitumour immunity. *Nature* **630**, 457–465 (2024). <https://doi.org/10.1038/s41586-024-07443-8>
7. Zhang, A.Q., Hostetler, A., Chen, L.E. *et al.* Universal redirection of CAR T cells against solid tumours via membrane-inserted ligands for the CAR. *Nat. Biomed. Eng* **7**, 1113–1128 (2023).
<https://doi.org/10.1038/s41551-023-01048-8>
8. Zhang, D.K.Y., Adu-Berchie, K., Iyer, S. *et al.* Enhancing CAR-T cell functionality in a patient-specific manner. *Nat Commun* **14**, 506 (2023). <https://doi.org/10.1038/s41467-023-36126-7>
9. Jennifer N. Brudno, James N. Kochenderfer; Toxicities of chimeric antigen receptor T cells: recognition and management. *Blood* 2016; **127** (26): 3321–3330. doi: <https://doi.org/10.1182/blood-2016-04-703751>
10. Norelli, M., Camisa, B., Barbiera, G. *et al.* Monocyte-derived IL-1 and IL-6 are differentially required for cytokine-release syndrome and neurotoxicity due to CAR T cells. *Nat Med* **24**, 739–748 (2018).
<https://doi.org/10.1038/s41591-018-0036-4>.
11. Giavridis, T., van der Stegen, S.J.C., Eyquem, J. *et al.* CAR T cell–induced cytokine release syndrome is mediated by macrophages and abated by IL-1 blockade. *Nat Med* **24**, 731–738 (2018).
<https://doi.org/10.1038/s41591-018-0041-7>.
12. Zhu, E., Yu, J., Li, YR. *et al.* Biomimetic cell stimulation with a graphene oxide antigen-presenting platform for developing T cell-based therapies. *Nat. Nanotechnol.* **19**, 1914–1922 (2024).
<https://doi.org/10.1038/s41565-024-01781-4>
13. Zhang, D.K.Y., Brockman, J.M., Adu-Berchie, K. *et al.* Subcutaneous biodegradable scaffolds for restimulating the antitumour activity of pre-administered CAR-T cells. *Nat. Biomed. Eng* **9**, 268–278 (2025). <https://doi.org/10.1038/s41551-024-01216-4>

Point-by-Point Response to Reviewer's Comments (Zheng et al. formerly manuscript NCOMMS-25-30476A)

For ease of reading, the reviewer's comments are in regular typeface, our responses are highlighted in italics

Point by point reply to Reviewer #2 (Remarks to the Author):

The authors have provided a variety of new/altered data and analysis in response to the original reviews. Unfortunately, however, they did not respond adequately to the main technical concern from the original review - a comparison of the optimized Dynabeads and optimized dextran nanoparticles is required to establish if this new platform provides a true advantage. The experiment here is quite straightforward - repeat previous experiments with a range of Dynabead:cell ratios, similarly to how the studies were performed with their particles, but the authors did not perform.

Response: We thank the reviewer for emphasizing the importance of testing optimized Dynabeads™ conditions. To address this concern, we conducted additional experiments evaluating a range of Dynabeads™:T-cell ratios (0.3:1, 1:1, 3:1, and 5:1), alongside a control group without beads. We have confirmed that a 3:1 Dynabeads™:cell ratio represents the optimized condition for a fair comparison with T-Expand, based on the following factors:

(1) Expansion efficiency:

All Dynabeads™-stimulated groups achieved measurable T-cell expansion by day 10 compared to the unstimulated control. Among the tested conditions, ratios of 3:1 and 5:1 yielded the highest cell numbers (Supplementary Fig. 16a, b in revised supplementary materials and added below), reaching levels consistent with those typically required for clinical-scale CAR T-cell manufacture.

(2) CAR transduction and expansion:

Following lentiviral transduction of CD19 CAR, both the transduction efficiency and expansion fold change were maximal at Dynabeads™:cell ratios of 3:1 and 5:1 (Supplementary Fig. 16 c, d in revised supplementary materials and added below). However, given that the 5:1 condition produced T cells with marginally higher exhaustion marker expression, we identified the 3:1 ratio as the optimal balance between proliferation and phenotypic preservation.

(3) Phenotype analysis:

Flow cytometry profiling revealed that increasing the Dynabeads™:cell ratio above 1:1 resulted, resulted in increased proportion of Terminally differentiated effector memory (TEMRA) T cells accompanied by a gradual decline in the effector-memory (TEM) subset (Supplementary Fig. 16e in

revised supplementary materials and added below). Also, the higher bead-to-cell ratios slightly elevated the expression of exhaustion-associated markers (e.g., PD-1, LAG-3, TIGIT) (*Supplementary Fig. 16f, g, h in revised supplementary materials and added below*).

Collectively, these results confirm that the standard 3:1 Dynabeads™:T-cell ratio represents the optimal condition for robust clinical T-cell activation and CAR T-cell expansion. This finding validates that the comparison in our study that between T-Expand and Dynabeads™ using this optimized (and provide recommended) dose.

We have added the following sentence in the results section: *Optimization experiments revealed that a Dynabeads™:T-cell ratio of 3:1 yielded the most efficient CAR-T cell production (Supplementary Fig. 16a-h). Consequently, this ratio was selected as the benchmark condition for comparison with T-Expand.*

And included the new Supplementary Fig. 16, as shown below.

Supplementary Fig. 16 T cells activated and transduced with CD19 CAR and expanded for 10 days using different Dynabeads™ to cell ratio. (a) Fold expansion of T cell from Day 0 to Day 10. (b) Total absolute cell numbers on Day 10 of expansion (c) Percentage of CAR T cells in total CD3+ T cell population analyzed using flowcytometry (d) Total number of CAR+ T cells on Day 10 (e) Phenotypic distribution of T-cells based on CCR7 and CD45RA expression, showing TN (CD45RA⁺CCR7⁺), TCM (CD45RA⁻CCR7⁺), TEM (CD45RA⁻CCR7⁻), and TEMRA (CD45RA⁺CCR7⁻) subsets. (f-h) T cells from Day 10 stained for exhaustion markers (f) LAG3 (g) TIGIT (h) PD1 and analyzed using flowcytometry.

Reviewer Report

Manuscript title: *Dextran-based T-cell expansion nanoparticles for manufacturing CAR T cells with augmented efficacy*

Authors: Tao Zheng et al.

Manuscript tracking number: NCOMMS-25-30476-T

The manuscript by Tao Zheng and colleagues presents a comprehensive and innovative study describing the development and evaluation of a novel dextran-based nanoparticle platform, termed T-Expand, for the *ex vivo* expansion of CAR T cells. According to the authors, this platform addresses key limitations of current artificial antigen-presenting cell (aAPC) systems such as Dynabeads™ and TransAct™, including overstimulation, poor scalability, and challenges in manufacturing.

In the first part of the study, the authors design and characterize the T-Expand nanoparticles, which feature a high surface density of anti-CD3 and anti-CD28 antibodies conjugated via click chemistry to the modifiable side chains of dextran. They claim that the nanoscale size and biocompatibility of T-Expand eliminate the need for post-expansion removal, simplifying the manufacturing process and improving scalability.

In the second part of the study, the functional characteristics of CAR T cells expanded using T-Expand are assessed. The results suggest that these cells possess enhanced phenotypic and functional attributes, including a greater proportion of CD27⁺ and CD28⁺ effector memory T cells, reduced expression of exhaustion markers, and improved anti-tumor efficacy both *in vitro* and *in vivo*. Furthermore, bulk RNA sequencing reveals upregulation of effector genes, including cytokines, and chemokine receptors implicated in T cell trafficking and immune modulation. The authors propose that these transcriptional changes correlate with the superior *in vivo* persistence and anti-tumor performance observed in xenograft lymphoma models, where T-Expand-expanded CAR T cells outperformed higher-dose Dynabeads™-expanded cells.

This is a timely and overall well-executed study that introduces a promising new platform for CAR T-cell manufacturing. It represents a meaningful contribution to the field of adoptive cell therapy, with the potential to improve therapeutic efficacy and streamline production. However, before publication, the authors should address the following major and minor concerns:

Major concerns

1. The authors report the optimization of anti-CD3 and anti-CD28 antibody ratios used for the functionalization of the T-Expand particles. However, no data are provided regarding the final amount of each antibody successfully conjugated to the particles. Since conjugation efficiency may differ from the input ratios, quantification (e.g., using ELISA or fluorophore-tagged antibodies) is important to clearly report the final antibody composition present on the T-Expand particles used in the experiments.
2. While the authors emphasize the biocompatibility of the T-Expand nanoparticles as a key feature, no clear experimental data are provided to support this claim. The presented T cell proliferation data do not address the concentration-dependent effects of the particles on the overall cell population. In this context, I recommend including a biocompatibility assessment, such as cytotoxicity or cell proliferation assays using T-Expand particles at varying concentrations on T cells and/or other relevant cell lines, to confirm the system's safety profile.
3. Did the authors use equivalent concentrations of Dynabeads™, TransAct™, and T-Expand particles, considering their differing physical and biochemical nature? Specifically, do these platforms contain comparable amounts of anti-CD3 and anti-CD28 antibodies? Given that T-Expand is a nanoparticle-based system, while Dynabeads™ and TransAct™ are microscale or polymer-based platforms, direct comparison may be confounded by differences in particle size, surface area, and antibody density. For example, was normalization performed based on the number of particles, the total antibody content, surface density of antibodies, or functional activity? The authors should clearly describe and justify the approach used to compare the different platforms and discuss whether the different concentrations used may lead to different results.

4. The authors claim that the particles do not require post-expansion removal due to their nanoscale size and biocompatibility. However, this assumes the particles remain stable and do not aggregate, accumulate, or degrade over time. I strongly recommend assessing the stability and integrity of T-Expand particles over relevant time points under physiological experimental conditions.
5. RNA-seq data indicate substantial upregulation of IFN- γ , pro-inflammatory cytokines (e.g., IL-6, IL-17), and other immune mediators in T-Expand-expanded CAR T cells. While these signatures may correlate with increased functionality, they also raise concerns about the potential for cytokine release syndrome (CRS) and immune effector cell-associated neurotoxicity syndrome (ICANS), two major clinical risks associated with CAR T-cell therapy. The authors should explicitly discuss these risks in the manuscript and consider whether the upregulated cytokine profiles observed for the T-Expand approach may pose translational safety concerns. In this context, it would be informative if the authors could report and discuss whether they observed any adverse reactions in their CAR T-cell experiments in mice.
6. Additional discussion is needed regarding the differing functional outcomes observed between the Dynabeads™ and T-Expand platforms. This is particularly important, as the two systems yield distinct clinical-relevant T cell activities and gene expression profiles. How do the authors explain the observed divergence in activity? Is the unique nature of the T-Expand nanoparticles, such as their size, flexibility, or nanoscale interaction dynamics, sufficient to account for these changes? Could the chemical composition of the dextran polymer itself contribute to altered T cell activation or downstream signaling? Alternatively, might these differences arise from distinct physicochemical properties of the particles, such as surface charge, stiffness, or antibody orientation and density? The manuscript currently lacks a detailed analysis and discussion regarding this point and a thoughtful discussion of these mechanisms would greatly strengthen the manuscript and support the claims made about the advantages of the T-Expand system over conventional aAPC platforms.

Other comments:

1. In lines 142–143, the authors state that T-Expand particles exhibit high uniformity in cryo-EM images. However, no quantitative analysis is presented from the cryoEM micrographs. Instead, the authors rely on NTA data showing a mean particle size of 199.8 ± 70.7 nm. Figure 1h shows a secondary peak at ~ 120 nm, suggesting the presence of a distinct subpopulation. The authors should discuss this heterogeneity and clarify whether the smaller peak represents unbound particles. If possible, quantify and report the proportion of each population.
2. Overall, the authors should provide more comprehensive descriptions of their experimental procedures, as the current level of detail is insufficient for reproducibility. For example, in lines 501–514, the section describing the formation of T-Expand particles for T cell activation and expansion lacks critical experimental information. This includes the concentrations of antibodies used, reaction volumes, buffer compositions, solvents, and technical details of the instrumentation employed. The authors are encouraged to thoroughly revise and complete all experimental sections. Specifically, the sources and references for the anti-CD3 and anti-CD28 antibodies should be clearly stated. Additionally, the origin of all cell lines used, along with their corresponding unique identifiers or commercial catalog numbers, should be provided in the methods to ensure clarity and traceability.